



# Carbon Burial in two Greenland Fjords: Exploring the Influence of Glacier Type on Organic Carbon Dynamics

Marius Buydens[1], Emil De Borger[1], Lorenz Meire[2,5], Samuel Bodé[3], Antonio Schirone[4], Karline Soetaert[5], Ann Vanreusel[1], Ulrike Braeckman[1,6]

[1]Marine Biology Research Group, Ghent University, Krijgslaan 281, S8 9000, Gent, Belgium
[2]Greenland Climate Research Centre, Greenland Institute of Natural Resources, Kivioq 2, 3900 Nuuk, Greenland
[3]Isotope Bioscience Laboratory (ISOFYS), Ghent University, Coupure Links 653, 9000 Ghent, Belgium
[4]ENEA, Department of Sustainability, Marine Environment Research Centre S. Teresa, Via Santa Teresa 1, 19032 Pozzuolo di Lerici, Italy
[5]Royal Netherlands Institute of Sea Research (NIOZ), Department of Estuarine and Delta Systems, Korringaweg 7, P.O. Box 140, 4401, NT, Yerseke, the Netherlands
[6]Institute of Natural Sciences, Operational Directorate Natural Environment, Vautierstraat 29, 1000, Brussels, Belgium

*Correspondence to*: Marius Buydens (marius.buydens@ugent.be)

**Abstract.** Fjord systems are crucial for the burial and long-term storage of organic carbon (OC), contributing significantly to
global blue carbon sequestration. Despite their importance, Greenland's fjords remain underrepresented in global carbon budgets, even though accelerated melt of the Ice Sheet alters these ecosystems through increased freshwater discharge and iceberg calving, ultimately leading to glaciers retreating inland. This study compares organic carbon burial rates (OCBRs) in two neighbouring Greenland fjords—Nuup Kangerlua, influenced by marine-terminating glaciers (MTGs), and Ameralik, dominated by land-terminating glaciers (LTGs)—to explore the effects of both types of glaciers on sediment carbon dynamics.
Since subglacial discharge-driven upwelling in Nuup Kangerlua (MTG) has been shown to support higher summer phytoplankton blooms, we expected higher sediment organic carbon content and burial in this MTG fjord. However, our observations show higher OC content in sediments of Ameralik's (LTG) outer and mid fjord section and a similar OCBR in both fjords. This unexpected finding may be linked to differences in pelagic grazing pressure, organic carbon transport, and sediment preservation mechanisms. The findings call for further research to unravel the complex interactions between primary
production, organic carbon transport, and preservation processes in different glacial fjord systems.

## 1 Introduction

Fjord systems play a crucial role in burial and long-term storage of organic carbon, contributing to approximately a tenth of the annual blue carbon burial (Smith et al., 2015). In the Northern hemisphere, carbon content and burial in fjord sediments have mainly been studied in Alaska (Cui et al., 2016a), Scotland and Ireland, (Smeaton et al., 2016; Smeaton and Austin, 2017;
Smeaton and Austin, 2019; Smeaton et al., 2021), Norway (Faust and Knies, 2019; Włodarska-Kowalczuk et al., 2019), Sweden (Placitu et al., 2024; Watts et al., 2024) and Svalbard (Kuliński et al., 2014; Koziorowska et al; 2018; Zaborska et al., 2018; Włodarska-Kowalczuk et al., 2019).



Nevertheless, despite the prevalence of fjords along Greenland's extensive coast, a notable gap remains in their representation in global carbon budgets (Smith et al., 2015). Moreover, Greenland harbours the only remaining Arctic ice sheet since the last

glacial period, which plays a key role in regulating Earth's climate and sea-level. The current accelerated melting of the Greenland Ice Sheet (King et al., 2020; Greene et al., 2024), driven by climate change, has far-reaching global implications and is altering fjord systems through increased freshwater discharge and iceberg calving (Calleja et al., 2017; Catania et al., 2020; Kanna et al., 2022).

Glaciers in polar regions either calve directly into the ocean (so called "marine-terminating glaciers", further referred to as MTGs) or terminate inland, discharging into lakes or the ocean via meltwater rivers ("land-terminating glaciers", LTGs). Fjords, inundated relict valleys carved out during previous glacial periods, often serve as channels through which these glaciers and meltwater rivers reach the ocean. Meltwater percolates down the cracks and crevices of glaciers to ultimately form sub-glacial rivers at their base (Chu, 2014). Since MTGs terminate in the ocean, this sub-glacial meltwater rises up from the bottom

of the glacier within the fjord basin entraining nutrients present in deeper water layers (Hopwood et al., 2020 and references therein). This upwelling water mass replenishes essential nutrients in the surface waters, crucial for sustaining phytoplankton proliferation beyond the initial spring bloom phase. This extended bloom, running into the summer months may potentially lead to increased organic carbon production within the fjord ecosystem (Kanna et al., 2022; Meire et al., 2023). Conversely, fjords receiving meltwater from LTGs lack this mechanism of upwelling, leading to a depletion of nutrients following the

spring bloom period, resulting in substantially lower primary production in summer (Meire et al., 2017, 2023). Consequently, the carbon dynamics in LTG-dominated fjords may differ significantly from those observed in MTG-dominated fjords.

An important characteristic of fjord systems that enhances their capacity as carbon sinks is an elevated sedimentation rate, driven by their proximity to glaciers and rivers, along with the steep terrain of their watersheds (Syvitski, 1987). However,

sedimentation rate alone is not the sole determinant of effective carbon burial (Bianchi et al., 2020). In general, the effectiveness of trapping organic carbon varies among fjords and depends on (1) the productivity of the fjord waters, particularly phytoplankton growth, as well as terrestrial vegetation growth in the catchment, both of which are influenced by climate (e.g. fjord categories described in Włodarska-Kowalczuk et al., 2019), (2) factors affecting the settlement of organic carbon (OC) to the fjord's bottom sediments (e.g. fjord geomorphology and current dynamics) (Gilbert et al., 2002; Faust and

Knies, 2019; Watts et al., 2024) and (3) factors limiting the degradation of settled OC, among which the refractory nature of OC (Koziorowska et al., 2015; Zaborska et al., 2018), sedimentation rate (Watts et al., 2024) and bottom water redox conditions (Hinojosa et al., 2014).

Findings from the limited number of biogeochemical studies focusing on Greenland fjords have sparked speculation that

enhanced primary production observed in MTG-dominated fjords, driven by the upwelling effect, may lead to increased carbon sequestration in fjord sediments compared to LTG-influenced fjord systems (Meire et al., 2017; Meire et al., 2023; Stuart-Lee



et al., 2023). However, there is limited data from Arctic fjords to test this hypothesis. In Svalbard, a lower OC content has been observed in the surface sediments of a LTG-fed fjord compared to two MTG-impacted fjord systems (Laufer-Meiser et al., 2021). In contrast, another study conducted in Svalbard reported a higher OC content in the surface sediments of a LTG-
compared to a MTG-influenced fjord (Koziorowska et al., 2015). While the first study ascribed the observed pattern to the glacier-driven upwelling effect, the second study attributed the higher OC content to the higher proportion of terrestrially-derived organic matter versus the more degradable marine organic matter. A study comparing organic carbon burial rates (OCBR) in Arctic fjords stated that high Arctic fjords with limited glacial activity and a short phytoplankton growth period sequester lower amounts of carbon in the sediments compared to Arctic fjords with "active" glaciers and a relatively longer
phytoplankton growth period (Włodarska-Kowalczuk et al., 2019).

This study aims to improve our understanding of carbon burial processes in Greenland fjord systems and provide insights that may refine estimates of their contribution to carbon burial at regional scales. In addition, we seek to gain insights in the influence of different types of Greenland fjord systems, more specifically in terms of MTG or LTG discharge influence. This
knowledge is crucial for developing a comprehensive understanding of how climate change may impact the long-term carbon storage capacity of Greenland fjord systems and the potential related feedback effects on global climate systems.

Therefore, we compared carbon storage and burial in two neighbouring, sub-Arctic fjord systems which both feature a sill at their entrance and are subjected to similar offshore currents and similar geology in their catchments, but have a different glacier
influence (MTG-dominated vs. LTG-dominated fjords).

## 2 Materials and methods

### 2.1 Study area

The two fjord systems are situated in the sub-Arctic coastal region of Southwest Greenland. Covering an area of 2,013 km$^2$, Nuup Kangerlua (formerly known as Godthåbsfjord) forms, with its many side branches, the largest fjord system of West
Greenland (Mortensen et al., 2018). The main branch is ~190 km long. Three marine-terminating glaciers and three meltwater rivers discharge into the fjord (Mortensen et al., 2011; Fig. 1). The land-terminating glaciers release $7.5 \pm 2.1$ km$^3$ yr$^{-1}$ of freshwater into the fjord system, while the marine-terminating glaciers supply $18.4 \pm 5.8$ km$^3$ yr$^{-1}$ of freshwater in addition to $7$–$10$ km$^3$ yr$^{-1}$ of solid ice discharge (Van As et al., 2014; Langen et al., 2015). The seafloor morphology comprises two consecutive sills at the fjord entrance and a third sill located in the inner fjord area in front of the termini of the two
innermost MTGs (Mortensen et al., 2011; Fig. 1). Inflow of dense coastal waters renews basin water masses in the main fjord basin usually from November until April (Mortensen et al., 2011, 2014, 2018). Bottom water temperatures are between 1.5 and 2°C (Mortensen et al., 2011).





Ameralik is situated south of Nuup Kangerlua, and has a length of around 75 km and a surface area of 400 km$^2$ (Stuart-Lee et

al., 2023). The fjord receives most of the freshwater runoff from a meltwater river (Naujat Kuat) draining an inland glacier.

Overeem et al. (2015) measured in 2012 a discharge of 0.78 km$^3$ yr$^{-1}$ of Naujat Kuat into the fjord. A large sill is situated at

the mouth of Ameralik and rises to 110 m water depth (Stuart-Lee et al., 2021). Being more than twice as shallow compared

to the entrance sills in Nuup Kangerlua, the sill restricts inflow of relatively warmer and more saline sub-polar mode water

(SPMW), resulting in relatively lower bottom water temperatures of ~0–1°C (spring and summer 2019 data; Stuart Lee et al.,

2021). The seafloor geomorphology behind the sill consists of a series of basins with the deepest and more extensive basin

situated about 20 km inwards from the main sill. Within this basin, the bathymetry plummets to a water depth of approximately

730 m.

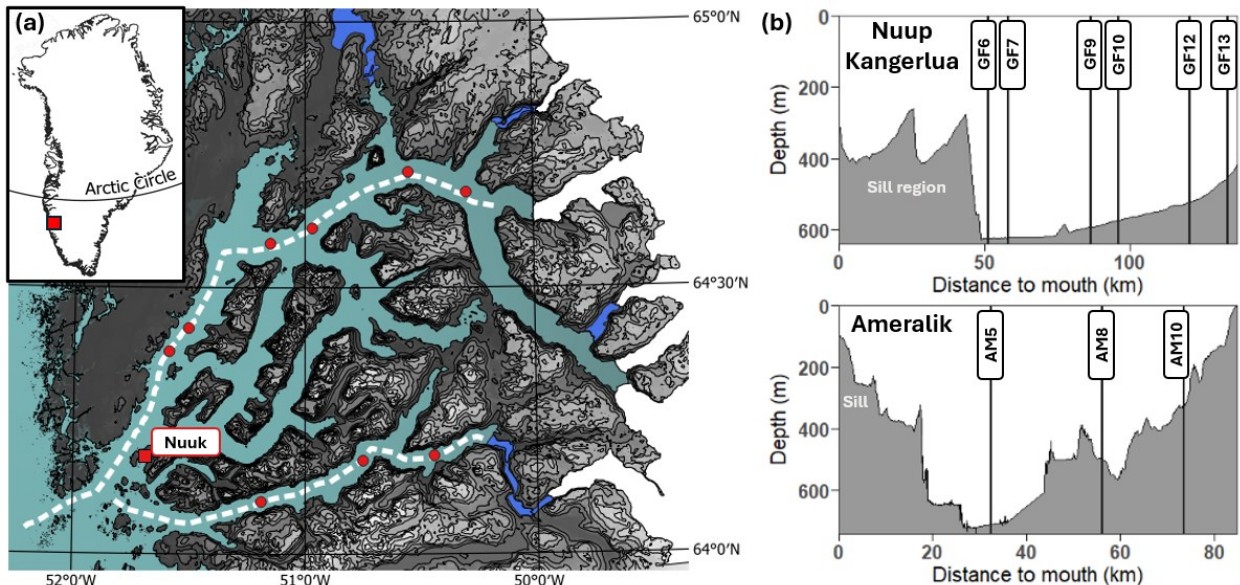

**Figure 1: A.** Red dots mark the locations of the sampled stations in Nuup Kangerlua (upper fjord; from mouth to head: GF6, GF7, GF9,
GF10, GF12 & GF13) and Ameralik (lowermost fjord; from mouth to head: AM5, AM8 & AM10). Areas indicated in blue represent the
main river valleys and deltas. Glaciers and Ice Sheet are indicated in white. White dashed lines shows the transects of the depth profiles. **B.**
Water depth profiles along-axis (white dashed lines) Nuup Kangerlua (top) and Ameralik (bottom).

**2.2 Sediment sampling**

Two field campaigns were organized, one in summer 2021 and one in spring 2022. Sediment samples were taken from the

research vessels *Polar Diver* (2021) and *Avataq* (2022). A UWITEC multicorer (UWITEC GmbH, Austria) was deployed to

sample the seafloor and included three core liners of 60 cm long with an inner diameter of 8.6 cm. Stations were located along

the main axis of both fjords (Fig. 1). No successful deployments could be carried out at the sill areas situated at the mouth area

of both fjords due to the high abundance of gravel. Sampled stations are therefore located behind the sills, within the fjord

basin. Although the mouth areas of both fjords could not be sampled, we divided each fjord basin into "outer,' "mid," and



"inner" sections for clarity. Throughout the text, the terms "outer," "mid," and "inner" refer to specific station locations. For Nuup Kangerlua, the "outer" area corresponds to stations GF6 and GF7, the "mid" area to GF9 and GF10, and the "inner" area to GF12 and GF13. In Ameralik, the "outer", "mid", and "inner" fjord areas correspond to stations AM5, AM8, and AM10, respectively.

### 2.2.1 Solid phase sampling

At each station, three deployments were carried out for granulometry, pigment, TOC and TN analysis (n = 3) and one deployment for porosity and $^{210}$Pb analysis (n = 1) and for stable isotope analysis of C and N (n = 1; in 2022). Due to the more exploratory approach of the 2021 campaign, less stations and parameters were sampled compared to 2022 (Table 1). The retrieved sediment was sliced into 1 cm thick slices down to 10 cm sediment depth. Sediment intended to derive sediment accumulation rates ($^{210}$Pb analysis) was further sliced beyond 10 cm in intervals of 2 cm until the end of the core (ranging from

10 to 44 cm sediment). All samples were stored at -20 °C, except for sediment samples intended for pigment analysis, which were stored at -80 °C.

**Table 1** Sampling dates, coordinates, water depth, and bottom water temperatures (BWT) of sampled stations in Nuup Kangerlua (Godthåbsfjord; GF) and Ameralik (AM).

| Station | Date(s) sampled | Longitude (N) | Latitude (W) | Depth (m) | BWT (°C) 2021 | 2022 |
|---------|-----------------|---------------|--------------|-----------|---------------|------|
| GF13 | 31/05/2022 | 64° 40.8 | 50° 17.3 | 476 | 1.4742 | 1.4135 |
| GF12 | 31/08/2021 20/05/2022 | 64° 42.9 | 50° 32.8 | 531 | 1.4135 | 1.347 |
| GF10 | 31/08/2021 | 64° 36.6 | 50° 57.5 | 570 | 1.3239 | 0.8074 |
| GF9 | 24/05/2022 | 64° 33.0 | 51° 0.9 | 626 | 1.228 | 0.6737 |
| GF7 | 01/09/2021 20/05/2022 | 64° 25.5 | 51° 3.4 | 630 | 1.2908 | 0.6387 |
| GF6 | 30/08/2021 | 64° 22.0 | 51° 0.4 | 630 | 1.2791 | 0.6185 |
| AM10 | 02/09/2021 18/05/2022 | 64° 11.0 | 50° 25.9 | 350 | 0.4943 | 0.4452 |
| AM8 | 18/05/2022 | 64° 10.4 | 50° 45.3 | 488 | 0.5925 | 0.5571 |
| AM5 | 03/09/2021 24/05/2022 | 64° 05.7 | 51° 11.3 | 730 | 0.5597 | 0.59 |






## 2.3 Solid phase sample analysis

Grain size distribution was determined on oven-dried samples (at 60 °C for 48 h) applying the laser diffraction method using a Malvern Mastersizer 2000 with Hydro 2000S module (0.02–2000 mm size range). Results were categorized in clay (<4 mm),

silt (4–63 mm), and sand (63–500 mm) fractions conform Wentworth scale classification (1922). Porosity was obtained by dividing the porewater volume by the wet sediment volume. Samples were dried and homogenized, then analyzed for total sedimentary carbon (TC) and total nitrogen (TN). After decalcification with 37 % HCl, total organic carbon (TOC) was also measured. All measurements were conducted using a Flash 2000 NC Sediment Analyzer (Interscience). From these data, the molar CN ratios were calculated and inorganic carbon (IC) was determined by subtracting TOC from TC. To investigate the

origin of the organic matter (see further), stable isotope composition ($\delta^{13}$C and $\delta^{15}$N) was measured with an elemental analyzer (Thermo Flash EA1112 elementalanalyzer) coupled to an isotope ratio mass spectrometer (Thermo Finnigan Delta V, IRMS). To explore how glacier type affects marine primary productivity and whether and how it is incorporated in the sediment, we additionally measured, for each sediment slice, the content of chlorophyll-a (Chl-a) and of its degradation products (pheophorbide-a, and pheophytin-a, pheophorbide-a like, and pheophytin-a like following Wright and Jeffrey (1997). These

different pigments were determined by the response factor of standard pigments as described by Van Heukelem and Thomas (2001). The ratio of Chl-a to Chloroplastic Pigment Equivalent (CPE, comprising the sum of all aforementioned pigments) was used as a proxy for the "freshness" of photosynthetically produced organic matter.

### 2.3.1 Calculation of marine organic carbon fraction

Stable isotope composition in addition to C:N ratios of settled organic matter in fjord sediments has been used in multiple

studies to estimate the proportion of marine versus terrestrially derived OM (Koziorowska et al., 2015; Smeaton & Austin, 2017; Faust and Knies, 2019). However, the use of solely stable isotopes can render an overestimation of marine OM as eroded and reburied fossil carbon from rocks (petrogenic carbon) display $\delta^{13}$C values within the range of recent marine OM masking a marine fossil provenance (Burdige, 2007; Cui et al., 2016b; Wang et al., 2024). The bedrock of the catchments of both fjords is predominantly made up of Precambrian orthogneisses, granodiorites and granites. Potential sources of petrogenic carbon

like meta-sedimentary rocks occur, but are rather rare in the catchment areas (<0.1 % of exposed bedrock) (Næraa et al., 2014). Therefore, it is reasonable to assume that the input of ancient marine carbon is likely to be limited. The fraction of OC derived from terrestrial C was calculated following the formula of Thornton and McManus (1994):

$$OC_{terrestrial} = \frac{\delta^{13}C_i - \delta^{13}C_M}{\delta^{13}C_T - \delta^{13}C_M} \qquad\qquad (1)$$

and

$$OC_{marine} = 1 - OC_{terrestrial}, \qquad\qquad (2)$$

where $\delta^{13}C_i$ represents the surface sediment values (0–2 cm) of $\delta^{13}C_{org}$ of each sample, $\delta^{13}C_M$ is the marine end-member and $\delta^{13}C_T$ is the terrestrial end-member. These end-members were adopted from Faust and Knies (2019): -19.3‰ and -26.5‰ vs.





V-PDB-LSVEC, for the marine and terrestrial end-member, respectively. These end-members were derived from Northern and Mid-Norway fjord sediments and agree with western Barents Sea sediments (Knies and Martinez, 2009; Faust and Knies, 2019) and Svalbard fjord sediments (Winkelman and Knies, 2005).

## 2.4 $^{210}$Pb and $^{137}$Cs analysis

Lead-210 dating of the sediment was done using HPGe gamma ray spectroscopy (diameter: 101.6 mm, height 134.9 mm, carbon-epoxy window, model BE5030-7500SL-RDC-4, Canberra, Asse, Belgium). The dried and grinded sediment samples were packed into aluminium tins with calibrated geometries of 35 ml, 60 ml or 120 ml, depending on the amount of dried sediment available, and left for > 21 days after sealing allowing ingrowth equilibration of the $^{226}$Ra with the proxies used to estimate its activity ($^{214}$Pb and $^{214}$Bi) (Brenner et al. 2004). When tins could not be filled entirely, the headspace was measured accurately, and an empirical model per geometry was used to correct for change in efficiency. The measurement of $^{210}$Pb activity was done using its 46.5-KeV gamma peak as described by Cutshall et al. (1983). The contribution of "supported" $^{210}$Pb was assessed by estimating the $^{226}$Ra activity from the average of the $^{214}$Pb (at 295.2 and 351.9 keV) and $^{214}$Bi (at 609.3 keV) activities. Supported $^{210}$Pb was then subtracted from the total $^{210}$Pb for each depth interval to determine "excess" $^{210}$Pb ($^{210}$Pb$_{xs}$). Additionally, $^{137}$Cs levels were determined through gamma spectroscopic measurement of its 661.7-KeV gamma peak.

### 2.4.1 Organic carbon burial rate constant

Log transformed $^{210}$Pb$_{xs}$ activities were plotted against the cumulative dry mass depth (g cm$^{-2}$) of the sediment per station. Sedimentation rates at stations GF9, GF7, AM8 and AM5 were determined using the constant rate of supply model (CSR) (Appleby, 2001), as a clear $^{137}$Cs peak was measured at these stations (Fig. A1). For the stations GF13, GF12, GF10, AM10 and GF6, the sediment mass accumulation rate (MAR, kg solids cm$^{-2}$ yr$^{-1}$) was derived from the slope of the linear regression according to the CF:CS model (Constant Flux:Constant Sedimentation) (Sanchez-Cabeza & Ruiz-Fernández, 2012). The bulk sediment accumulation rate (SAR, mm yr$^{-1}$) was calculated by dividing MAR by the average bulk density of the sediment per station. The organic carbon burial rate (OCBR) per station was determined using the MAR and the TOC content at the deepest sediment layer in common for all sediment cores (9–10 cm sediment interval depth). No bioturbated or mixed upper layer was observed in the profiles.

## 2.5 Statistical analysis

We examined differences between the two fjords and between stations in terms of sedimentary TOC and TN content, C/N ratio, Chl-a content and Chl-a:CPE ratio from the upper 2 cm sediment surface and the average of the 10 cm sediment column.



Data from summer (2021) and spring (2022) were combined and treated as replicates, as the difference between the two seasons was insignificant. As a consequence, stations GF12, GF10, GF7, AM10 and AM5 have six replicates since they were sampled

in both seasons, while the other stations have three replicates as those stations were only sampled during spring 2022 (Table 1). Statistical analyses were performed using one-way ANOVA. Welch's ANOVA was applied when variances were unequal, and the Kruskal-Wallis test was used when normality assumptions were violated. For significant results, post hoc comparisons were made using Tukey's test, Games-Howell test, or Dunn's test, depending on the initial method. Results are reported as means ± standard deviation. Statistical analyses were performed in R (R Core Team, 2023) using the car, rstatix and FSA

packages (Fox and Weisberg, 2019; Kassambara, 2023; Ogle et al., 2023).

## 3 Results

### 3.1 Solid phase parameters

The median grain size (d0.5) was situated in the silt fraction for all stations, though AM5, AM8 and the top 2 cm of GF7 displaying medium-sized silt, while all other stations are situated in the very fine to fine silt class (Fig. 2). In Nuup Kangerlua,

the median grain size (d0.5) shows a clear spatial pattern from the inner to the outer fjord (Fig. 2). At the inner stations (GF13 and GF12), the grain size remains relatively small (< 20 μm) and consistent with depth, reflecting a stable depositional environment dominated by fine particles. Moving to the mid-fjord stations (GF10 and GF9), there is a slight increase in grain size, though still within the fine-silt range, indicating limited hydrodynamic influence. At the outer stations (GF7 and GF6), the grain size increases further to medium-sized silt and varies with depth, suggesting the deposition of coarser sediments near

the fjord mouth due to stronger currents and more dynamic conditions.

In Ameralik, a similar trend is observed (Fig. 2). The inner station (AM10) shows small, uniform grain sizes comparable to those at the inner stations in Nuup Kangerlua. At the mid-fjord station (AM8), the grain size increases slightly, reflecting a subtle shift in depositional energy. The outer station (AM5) exhibits the largest grain sizes, with noticeable variability at depth, indicating more pronounced hydrodynamic sorting and energy fluctuations in this area.

Porosity and dry density fluctuated along the sediment depth gradient without a clear trend, except for station GF10, where porosity decreased and dry density increased with depth.




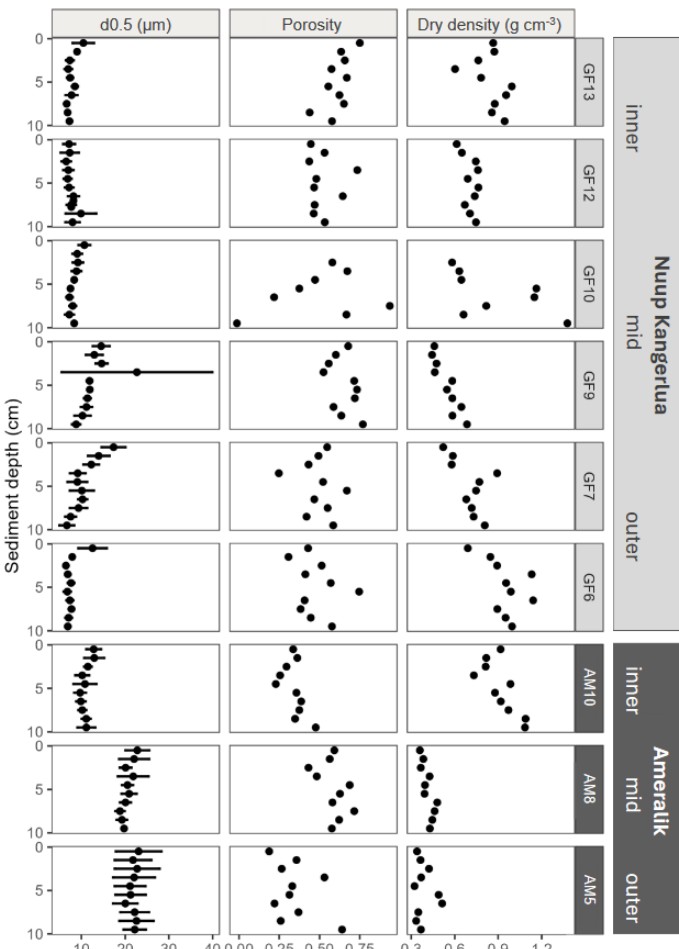

**Figure 2:** Sediment profiles of median grain size (μm), porosity and dry density (g cm⁻³) of Nuup Kangerlua (GF stations) and Ameralik (AM stations). Error bars represent SD (n = 3 for GF6, GF9, GF13 and AM8 and n = 6 for GF7, GF10, GF12, AM5 and AM10). Only one replicate for porosity and dry density.

In Ameralik, we observed a distinct increasing trend in the surface 2 cm sediments from the inner fjord to the mid-fjord stations for TOC, TN, Chl-a content, and the Chl-a:CPE ratio. The only exception was the C:N ratio, which decreased from the inner
to the mid-fjord, and then remained relatively constant (Fig. 3). In Nuup Kangerlua, the pattern was more variable. TOC, TN, and Chl-a content rose from the inner fjord towards the mid-fjord stations, peaking at GF7 and GF9, but then declined at GF6, near the outer fjord (Fig. 3). Unlike Ameralik, the Chl-a:CPE ratio in Nuup Kangerlua showed fluctuations along the fjord axis, without a consistent trend. Overall, station AM5, located in the deepest part of the main basin of Ameralik, displayed the highest (Welch's ANOVA, $p<0.05$) Chl-a (16.4±2.0 μg g⁻¹ DM) and CPE (45.9±7.1 μg g⁻¹ DM) content, as well as the highest
Chl-a:CPE ratios (0.36±0.04) of the top 2 cm surface sediments compared to all other sampled stations of both fjords (Fig. 3).





In addition, both outer and mid stations of Ameralik displayed the highest TOC values (AM5: 2.1±1.5 %; AM8:1.6±0.1 %) within the upper 2 cm sediment, which were significantly higher (Welch's ANOVA, $p<0.05$) than those observed in inner fjord station AM10 and all stations in Nuup Kangerlua (values ranging from 0.1 to 1.3 %).



**Figure 3:** Vertical sediment profiles depicting TOC and TN (%), molar C:N ratios and Chl-a (µg/g DM) of the upper 10 cm sediment of Nuup Kangerlua (GF stations) and Ameralik (AM stations). Error bars represent SD (n = 3 for GF6, GF9, GF13 and AM8 and n = 6 for GF7, GF10, GF12, AM5 and AM10).






Apart from GF13, the organic carbon in all stations within both fjords displayed δ¹³C values characteristic for marine algae (-22.4 to -20.7 ‰). While the δ¹³C value fluctuated widely at GF13 ranging from -26.3 to -23.8 ‰, while δ¹⁵N consistently increased with sediment depth from 5.7 to 12.2‰ (Fig. A2). The more depleted δ¹³C values at this station suggest relatively more mixing with terrestrial OM, while δ¹⁵N levels are more indicative of a marine origin, since values are well above 1 ‰

and no intensive agriculture exists in the region (Harris and Elliot, 2019; Fig. 4). Levels of δ¹³C in the upper 2 cm slightly increased from inner to outer stations within Nuup Kangerlua indicating an increasing marine influence in terms of organic carbon composition towards the outer fjord area. In Ameralik, the least depleted δ¹³C signatures were observed in AM8.

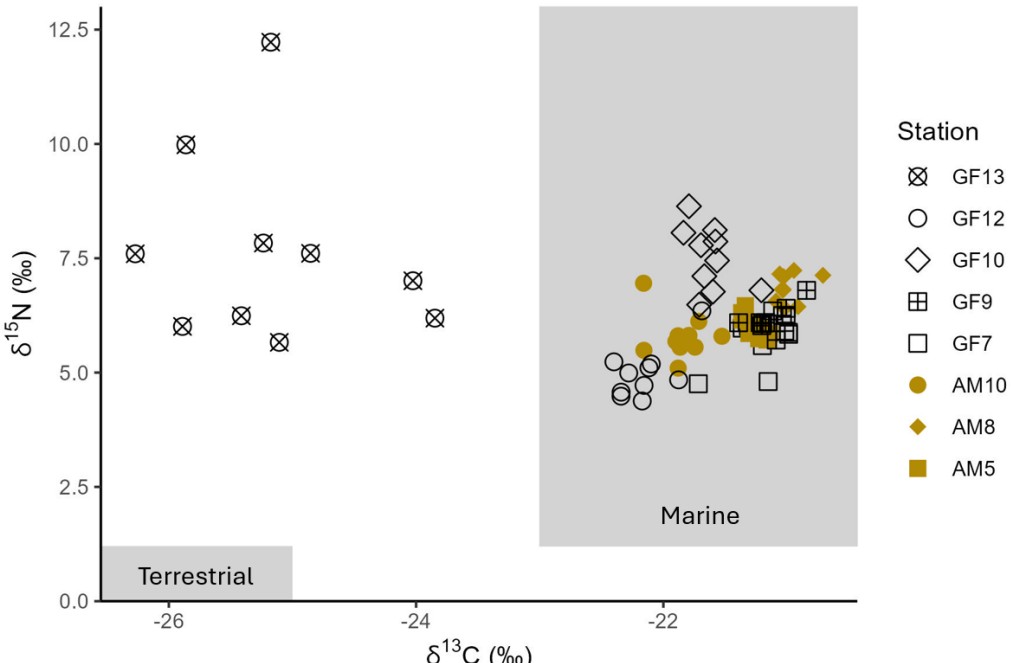

**Figure 4:** δ¹³C (‰ deviations from V-PDB) values plotted against δ¹⁵N (‰ deviations from air) values of the POM present in the sediment for the different station of Ameralik (colored) and Nuup Kangerlua (open symbols). Typical marine and terrestrial ranges of δ¹³C and δ¹⁵N are indicated with rectangles following Zaborska et al. (2018).

**3.2 Organic carbon burial rates**

Sediment mass (MAR) and volume accumulation rates (SAR) roughly showed an increasing trend towards the inner fjord in Nuup Kangerlua. In Ameralik, MAR and SAR are also higher in the inner compared to the outer station, with minimum values





in the mid station (Table 2). Burial rates of organic carbon increased towards the fjord head in Nuup Kangerlua until mid-fjord station GF10 where it reached the maximum observed rate (29.4 g OC m$^{-2}$ yr$^{-1}$). Following station GF12 revealed a marked

drop in OCBR (9.6 g OC m$^{-2}$ yr$^{-1}$), whereafter high OCBR reappear in GF13 (27.5 g OC m$^{-2}$ yr$^{-1}$). In Ameralik, an opposite pattern unfolded with maximum OCBR in AM5 (24.7 g OC m$^{-2}$ yr$^{-1}$) and minimum rate in inner fjord AM10 (9.9 g OC m$^{-2}$ yr$^{-1}$).

**Table 2.** Mass sediment accumulation rate (MAR), bulk sediment accumulation rate (SAR) and organic carbon burial rate (OCBR) per station. Stations are situated from mid fjord towards the head (inner): "GF" denotes Nuup Kangerlua and "AM" Ameralik.

| Station | MAR (kg m$^{-2}$ yr$^{-1}$) | SAR (mm yr$^{-1}$) | OCBR (g m$^{-2}$ yr$^{-1}$) |
|---|---|---|---|
| GF13 | 14.1 ± 3.5 | 15.0 ± 3.7 | 27.5 ± 8.3 |
| GF12 | 5.9 ± 1.0 | 7.1 ± 1.2 | 9.6 ± 1.7 |
| GF10 | 7.0 ± 0.1 | 8.3 ± 1.1 | 29.4 ± 4.0 |
| GF9 | 3.4 ± 0.1 | 5.1 ± 1.4 | 19.1 ± 5.2 |
| GF7 | 2.6 ± 0.1 | 3.5 ± 1.3 | 6.5 ± 2.7 |
| GF6 | 1.8 ± 0.1 | 1.9 ± 0.8 | 1.5 ± 0.6 |
| AM10 | 4.0 ± 2.8 | 5.2 ± 2.0 | 13.1 ± 5.0 |
| AM8 | 1.1 ± 0.0 | 2.4 ± 0.0 | 17.2 ± 0.3 |
| AM5 | 1.1 ± 0.0 | 2.9 ± 0.3 | 23.9 ± 2.3 |

## 255    4 Discussion

The OC content in the sediments of Nuup Kangerlua and Ameralik fall within the range of other (sub) Arctic fjord sediments (Fig. 5A). In addition, the OCBR values are representative for Arctic fjords (Włodarska-Kowalczuk et al., 2019). In terms of fresh organic matter, we found an average Chl-a content in Nuup Kangerlua's sediments which was slightly below the typical range observed in other North Atlantic fjords (Włodarska-Kowalczuk et al., 2019). In contrast, Ameralik exhibited an average



Chl-a concentration nearly three times higher than the maximum values reported for Svalbard fjords (Włodarska-Kowalczuk et al., 2019). This elevated average is largely driven by the exceptionally high Chl-a content observed at station AM5.

We expected higher OC content and OCBRs in Nuup Kangerlua compared to Ameralik, as MTGs present in Nuup Kangerlua increase nutrient upwelling, allowing primary productivity to extend over longer periods. Indeed, at the start of the productive season (April, May), Stuart-Lee et al. (2023) and Meire et al. (2023) noted comparable primary productivity in Nuup Kangerlua

and Ameralik. Yet, with increasing meltwater discharge, a summer bloom was observed in Nuup Kangerlua which led to a greater overall phytoplankton biomass compared to Ameralik (Stuart-Lee et al., 2023; Meire et al- 2023). Surprisingly, we found a higher OC content in sediments of outer and mid fjord stations AM5 and AM8 in Ameralik compared to Nuup Kangerlua. These findings are supported by observations from a gravity core sampled nearby station AM5, which also revealed similar elevated carbon content in the sediment (Møller et al., 2006).

When comparing datasets from other fjords (Thamdrup et al., 2007; Cui et al., 2016b; Faust and Knies, 2019; Włodarska-Kowalczuk et al., 2019; Laufer-Meiser et al., 2021), LTG systems and non-glaciated regions appear to have surface sediments with OC content comparable to those in MTG systems (Fig. 5A). This suggests that factors beyond glacial influence play a significant role in controlling the degree of benthic-pelagic coupling. Specifically, the presence of MTGs does not inherently result in higher OC accumulation within sediments compared to systems without subglacial upwelling. However, variations in

MARs may dilute OC concentrations with inorganic material, potentially skewing these observations. Additionally, higher TOC in surface sediments does not automatically equate to more efficient OC burial.

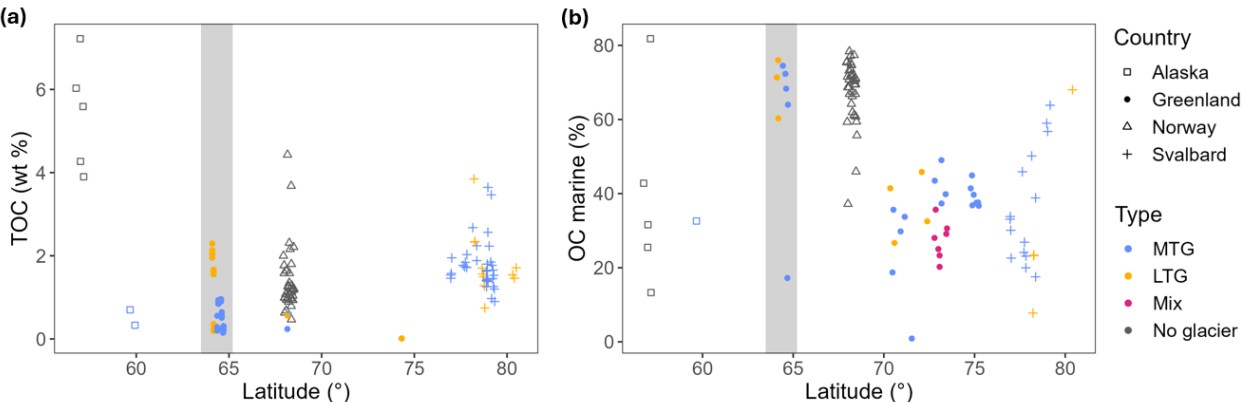

**Figure 5: A.** TOC content of surface sediments along latitude. **B.** Fraction of TOC of marine origin along latitude. Both figures display data from fjords located in high latitude countries: Alaska, Greenland, Norway and Svalbard. The grey band constraints the Greenland fjords investigated in this study. Data indicated in blue and red represent Marine terminating-glacier (MTG) and land terminating-glacier (LTG)-influenced fjord systems, respectively. The mixed type represents fjords where the dominance of MTG(s) vs LTG(s) on the fjord's hydrology could not be differentiated



from literature or satellite images. Fjord data illustrated in grey represent fjords hosting no major glaciers within their catchments. Both graphs were created following and updating the example of Faust and Knies (2019).

In what follows, we try to substantiate these apparent contradictory results by considering organic carbon, lability, the biogeochemical context, and variations in the export mechanisms between different fjords, Furthermore, recent findings on pelagic grazing pressures in both fjords highlight the role of food-web dynamics in carbon burial dynamics.


## 4.1 Enhanced OC preservation

An important clue in resolving the observed patterns can be found in the deepest part of Ameralik's basin. There, at station AM5, we measured a higher Chl-a content combined with higher Chl-a:CPE ratios compared to Nuup Kangerlua, which points to an enhanced preservation of fresh organic matter within these sediments. The Chl-a content remains elevated throughout the entire 10 cm sediment profile and is consistent between spring and summer data. A difference in timing of the onset of the

phytoplankton bloom between the two fjords, as previously observed (Stuart-Lee et al., 2023), could have led to an earlier build-up of pigments at the seafloor of Ameralik compared to Nuup Kangerlua at the time of sampling. However, the relatively elevated values throughout the 10 cm sediment profiles and the consistency between spring and summer data exclude such sampling bias. In Svalbard, Koziorowska et al. (2015) also observed higher OC content in the surface sediments of a LTG-

influenced fjord versus a MTG-impacted fjord. The LTG-fed fjord appeared to receive a higher fraction of terrestrial OC, which tends to be more resistant against degradation compared to marine OC (Koziorowska et al., 2015). Yet, in our case, the sediment stable isotope composition and C:N ratios of both fjords reflect OC of predominantly marine origin in both fjords, likely due to the limited vegetation and similar catchment geology consisting of orthogneisses, granodiorites and granites rather than organic-rich sedimentary rocks (Næraa et al., 2014) (Fig. 4; Fig. 5B). An exception is inner station GF13 in Nuup

Kangerlua, which displayed $\delta^{15}N$ and C:N ratios of marine signature, though depleted $\delta^{13}C$ values indicative of a terrestrial provenance.

Therefore, if a different degree in organic matter preservation occurs between the two fjords, it has to be linked to environmental conditions rather than the nature of the OC itself. Metal shielding, particularly the association of OC with iron or manganese, may be another factor influencing the preservation of sedimentary OC. Reactive iron minerals such as iron- and

manganese(oxyhydr)oxides preferentially bind to marine organic matter, forming complexes that enhance the protection of OC from microbial degradation and remineralization (Faust et al., 2022; Moore et al., 2023). Such carbon-iron and -manganese interactions have been found to persist over geological timescales, thereby enhancing long-term carbon sequestration (Faust



et al., 2021; Moore et al., 2023; Wang et al., 2024). Indeed, the Greenland Ice Sheet acts as a major source of Fe and Mn to its surrounding fjords through both subglacial and glacial river discharge (Bhatia et al., 2013; Hawkings et al., 2014; 2020).

Studies conducted in Nuup Kangerlua and Ameralik revealed that both fjords receive substantial amounts of Fe and Mn from (sub)glacial river inflow, which appears to be captured within the fjord basin rather than being exported offshore (Hopwood et al., 2016; Krause et al., 2021; van Genuchten 2021; 2022). Higher Fe and Mn concentrations were measured in the surface waters of inner Ameralik compared to Nuup Kangerlua, but similar concentrations appeared towards the outer area of both fjords (Krause et al., 2021; van Genuchten 2022). However, no information is available on the concentration of solid Fe and

Mn particles or derived complexes in the basin sediments. Moreover, differences in the balance and interplay between sedimentation rate, influx of organic matter and Fe and Mn species, the reactivity of Fe and Mn and the depth of sulphate reduction can play a role in differences in preservation of OC (Wehrmann et al. 2014; Michaud et al., 2020; Herbert et al., 2021; Laufer-Meiser et al., 2021). This complexity, however, goes beyond the scope of this study. Hence, it is not clear if metal shielding plays a role in elevated OC and Chl-a content in Ameralik's deep basin sediments.

The distinct geomorphology of Ameralik and Nuup Kangerlua, particularly their differing sill depths, likely shapes bottom water temperatures and may influence organic matter preservation within the fjords. Both fjords have no anoxic deep water masses and bottom water renewal occurs every one to two years (Mortensen, 2011; Stuart-Lee et al., 2021), but bottom water temperature differs. Ameralik's shallower sill depth (~110 m) compared to Nuup Kangerlua (~200 m) restricts the inflow of warmer, saltier coastal waters (Stuart-Lee et al., 2021). As a result, Ameralik's deep waters (below 400 m) are around zero

degrees and about one to two degrees colder than in Nuup Kangerlua (Stuart-Lee et al., 2021), which may cause the observed higher pigment preservation in AM5.

## 4.2 OC transport dynamics

In addition to potential differences in organic matter preservation, lateral transport can also play a role in shaping the spatial distribution of OC across the seafloor. In Nuup Kangerlua, a weak along-fjord gradient in sedimentary TOC, TN, and Chl-a content suggests a dynamic current regime that may facilitate OC redistribution. Meire et al. (2023) reported a threefold higher primary production rate at station GF10 compared to AM10 due to the summer bloom. Yet, we observe that this higher productivity did not translate into significant differences in sedimentary Chl-a and TOC content between these stations.

Although it must be noted that both parameters were relatively higher (although not significantly) at GF10 compared to AM10 and that TOC and Chl-a content at GF10 may have been diluted by the observed higher MAR.

Nuup Kangerlua's estuarine and subglacial circulations, which become most active during the melt season, may enhance OC export from the inner fjord to outer areas (Mortensen et al., 2011; 2014; Juul-Pedersen et al., 2015). Despite this potential for export in the surface waters, sediment trap data from Luostarinen et al. (2020) at GF10 (300 m depth) indicate a MAR and



TOC flux comparable to the calculated OCBR. This suggests either efficient preservation of OC settling beyond 300 m or contributions from an additional OC source.

     Tidal mixing at the mouths of both Nuup Kangerlua and Ameralik interacts with the sill topography, creating a density gradient that drives intermediate baroclinic circulation (Mortensen et al., 2011; Stuart-Lee et al., 2021). This circulation, characterized by out-fjord flow at depth and in-fjord flow near the surface, reintroduces nutrients to shallower layers, promoting

phytoplankton growth in the outer sections of both fjords (Stuart-Lee et al., 2023). In Ameralik, this local productivity likely accounts for the higher pigment and OC content observed in the outer fjord compared to the inner region. However, the absence of similar Chl-a and TOC trends at outer stations GF6 and GF7 in Nuup Kangerlua remains unexplained.

     Sørensen et al. (2015) suggested that the observed discrepancy between local primary production and the significantly higher POC export to the sediments in Kobbefjord—a small, glacier-free fjord located between Nuup Kangerlua and Ameralik—

could be due to an influx of OC from Nuup Kangerlua. Similarly, part of the OC produced in Nuup Kangerlua may be imported into Ameralik, contributing to increasing TOC and Chl-a content toward Ameralik's mouth. Both fjords exhibit estuarine and intermediate baroclinic circulation (Stuart-Lee et al., 2021), but OC transport efficiency appears greater in Nuup Kangerlua due to strong upwelling driven by subglacial discharge (Mortensen et al., 2014). Consequently, Ameralik may experience net OC import, with deep basin retention and settlement of OC potentially promoting enhanced preservation (Fig. 6). However, a

lack of current velocity data for Ameralik limits the ability to fully assess OC transport dynamics.

     Still, the observed slightly coarser grain-size fraction in Ameralik's outer and mid fjord stations may signal an input of material from the mouth area as this station is located too far from the glacier input to reveal coarser sediment compared to the inner part of the fjord. Sea Ice and icebergs which could transport coarser material further from the source are absent in the fjord nor is there a debris flow apparent from the grain-size and $^{210}$Pb profiles (Fig. A1). The coarser material may therefore originate

from the entrance sill indicating a more important deep water inflow compared to Nuup Kangerlua.





**Figure 6:** Schematic cross-sectional view of current regime and possible ways of phytoplankton or OC flow during summer in Nuup Kangerlua (**A**) and Ameralik (**B**) fjord systems. Tidal mixing above the sill area, estuarine circulation





and intermediate baroclinic circulation occurs in both fjord systems, while the presence of MTGs in Nuup Kangerlua drives subglacial circulation through subglacial discharge. Nutrients are brought to the euphotic zone via tidal mixing and subglacial circulation. Turbid plumes, indicative of suspended sediment and organic matter input from glacier discharge and river runoff, are represented by the shaded brown texture. Green arrows represent phytoplankton or OC transport and remineralization of organic carbon at the sediment-water interface. Station locations are marked along the fjords. The current dynamics illustrated for Nuup Kangerlua are based on Mortensen et al. (2018) and Stuart-Lee et al. (2023), while those for Ameralik are derived from Stuart-Lee et al. (2021; 2023).

## 4.3 Food web OC flow

As both fjords exhibit a high contribution of marine-derived OC compared to other Arctic fjord systems (Fig. 5B), the unexpectedly higher sediment OC content in the basin of Ameralik may be linked to differences in carbon cycling pathways, not only within the sediments, but potentially also above the seafloor. It is possible that the greater biomass and larger size class of phytoplankton in Nuup Kangerlua drive a more extensive and efficient food web (Meire et al., 2023; Stuart-Lee et al., 2023). As a result, a larger portion of OC is channelled into trophic transfer and remineralization, thereby reducing the amount

of OC reaching the seafloor compared to Ameralik (Fig 6).

As a consequence of the summer bloom, Nuup Kangerlua has a higher proportion of large herbivorous copepods, while smaller omnivorous species dominate in Ameralik (Stuart-Lee et al., 2024). However, despite these differences in primary producers and composition of zooplankton communities, Stuart-Lee et al. (2024) found no significant difference in zooplankton biomass between the two fjords during the entire melting season. This lack of difference may be influenced by the sampling methods

used, as the plankton net in that study was not optimal for capturing larger and more agile zooplankton such as krill (Stuart-Lee et al., 2024), which have been previously recorded in high abundances in the mid and inner part of the fjord (Agersted et al., 2011; Agersted & Nielsen, 2014). Another explanation may be that predation pressure exerts a control on the biomass of the larger and more zooplankton in Nuup Kangerlua (Stuart-Lee et al., 2024). Observations of higher Halibut landings in MTG-compared to LTG-influenced fjords in Greenland (Meire et al., 2017), as well as the importance of MTG fronts as foraging

spots for birds and mammals as observed in Svalbard (Lydersen et al., 2014; Urbanski et al., 2017; Vacquié-Garcia et al., 2018; Hamilton et al., 2019), suggest an important transfer of OC through various trophic levels in Nuup Kangerlua. The higher consumption of OC in the water column of Nuup Kangerlua might as such impact the vertical OC transfer to the sediment and result in lower OC content in the sediment of this fjord.



## 4.4 Organic carbon burial rates

Despite the higher organic carbon content observed in the outer and mid part of the LTG-fed fjord, organic carbon burial rates were similar in both fjords. The average OCBR in Ameralik was only slightly, but not significantly, higher ($16.5 \pm 1.7$ gC m$^{-2}$yr$^{-1}$) compared to Nuup Kangerlua ($14.1 \pm 1.6$ gC m$^{-2}$yr$^{-1}$) and fall within the range of Sub-Arctic fjords and Arctic fjords impacted by active glaciers (Włodarska-Kowalczuk et al., 2019). The higher MAR rates in Nuup Kangerlua result from the substantially higher discharge that three MTGs and three LTGs generate compared to the input of a single LTG in Ameralik. Apparently, there is no one-on-one relationship between glacier type and OCBR. Interestingly, despite known differences in pelagic primary production, carbon burial remains similar in both fjords, likely due to limited degradation of organic carbon in Ameralik relative to Nuup Kangerlua. Thus, while the amount and type of glaciers influence both primary production and MAR, the net effect on OCBR appears to be minimal.

## 5 Conclusion

This study provides new insights into carbon burial processes in two southwest Greenland fjords with a different type of glacier influence. Our findings point to complex processes at work regarding carbon burial as our data revealed a different pattern than generally assumed in literature (Hopwood et al., 2020). Our data show that primary production generates most of the organic matter ending up at the seabed sediments in two sub-Arctic fjords with similar metamorphic and igneous catchment geology. Despite the upwelling mechanism in place sustaining more primary production, this process does not seem to induce a higher OC burial in the seabed sediments of a MTG-impacted fjord compared to a LTG-fed fjord. In contrast, this upwelling could be responsible for an export of carbon out the fjord or promoting the transfer of carbon through a more extensive food-web. In that case, MTGs do function as carbon pumps where an important part of the produced OC is stored beyond the fjord basin sediments. However, differences in geomorphology or bottom water characteristics between the two fjords can also override the importance of the subglacial nutrient supply.

Our findings highlight the importance of investigating both the pelagic as benthic compartment of Greenland fjord systems, which are understudied and underrepresented in global carbon budgets compared to other regions. Although this study advances our understanding of the carbon dynamics in Greenland fjords, several unresolved questions remain. For example, differences in diagenetic processes between MTG- and LTG-influenced fjords, along with the role of physical circulation patterns in redistributing OC, require further investigation. Additionally, the potential for complex food webs and higher trophic interactions in MTG fjords to influence carbon sequestration deserves more attention.

Understanding the driving mechanisms of OCBR in fjord systems is essential to predict the impact of climate change on OC sequestration as MTGs evolve to LTGs. The similar OCBR observed between systems suggests that the retreat of MTGs from fjords may not necessarily reduce carbon burial, as new conditions influencing OCBR will emerge. Nevertheless, when assessing the impact of climate change on OC burial budgets, it is crucial to consider the fate of OC produced within the fjord.



**Appendix A**

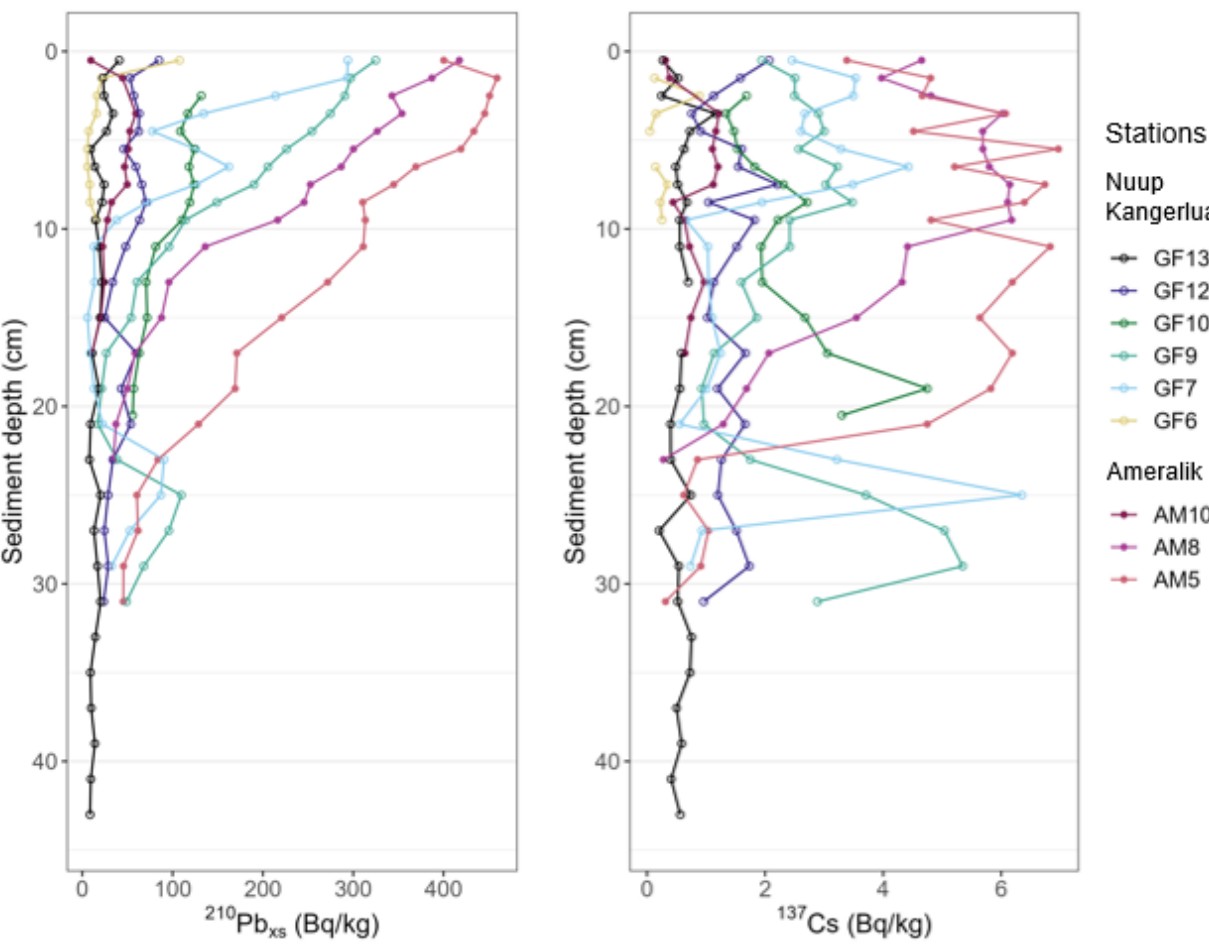

**Figure: A1.** Excess $^{210}$Pb and $^{137}$Cs profiles of Nuup Kangerlua stations (GF13, GF12, GF10, GF9, GF7 and GF6) and
Ameralik stations (AM10, AM8 and AM5).





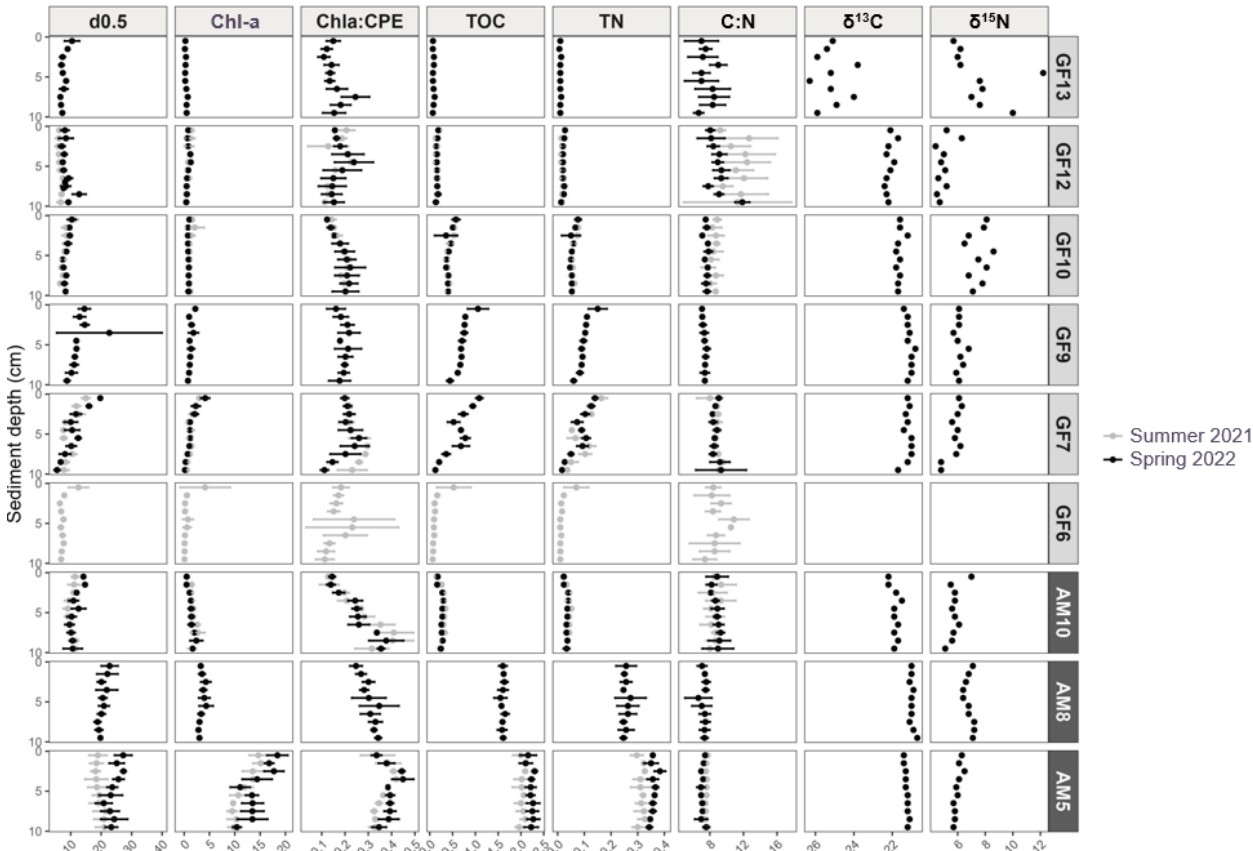

**Figure A2:** Vertical sediment profiles depicting average median grain size (μm), Chl-a content (μg g$^{-1}$ DM), Chl-a:CPE ratio, TOC and TN (%), and molar C:N ratios, and single core values of δ$^{13}$C (‰), δ$^{15}$N (‰), porosity and dry density (g cm$^{-3}$) of the upper 10 cm sediment of Nuup Kangerlua (stations GF13, GF12, GF10, GF9, GF7 and GF6) and Ameralik (stations AM10, AM8, AM5). Error bars represent SE (n = 3). Orange and black colors represent end of summer 2021 and spring 2022, respectively. Data from the two seasons is available for stations GF12, GF7, AM10 and AM5.

**Author contribution**

LM, AVR, KS and UB acquired funding for the research project and developed the overall research objectives. LM, MB, UB, AVR, KS and EDB contributed during the field work. SB supervised and carried out lab analyses of Pb $^{210}$ and Cs$^{137}$. MB conducted formal analysis and AS assisted in MAR calculations and interpretation. MB prepared the original draft and all authors reviewed the manuscript.

**Competing interests**

The authors declare that they have no conflict of interest.



**Acknowledgements**

Our sincere thanks to the Greenland Institute of Natural Resources for providing access to lab facilities and the research vessel *Avataq*, as well as accommodations during fieldwork. Special appreciation goes to Captain Peter Rosvig Pedersen and the crew of the *Polar Diver* for their assistance during field sampling. We thank MSc students Charles Makio, Marianne Lollevier
and Tran Manh Quan for their contributions to sample processing and preparation. We acknowledge Bart Beuselinck and Bruno Vlaeminck (Ghent University, Marine Biology Research Group) and Peter Van Breughel (NIOZ) for sample analysis. OpenAI's ChatGPT was used to check for grammar and improve the readability of the manuscript.

**Financial support**

This work is part of the IMAGIN project, funded by Research Foundation-Flanders (FWO) (grant no 3G043120). The research
leading to results presented in this publication was carried out with infrastructure funded by EMBRC Belgium - FWO international research infrastructure *I001621N*.

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
