# Peer review of "Carbon Burial in two Greenland Fjords: Exploring the Influence of Glacier Type on Organic Carbon Dynamics"

_EGUsphere, 2025_

## Author Comment (AC1)

We are grateful to you for the time you invested in providing detailed and helpful feedback. Your comments have been invaluable in refining and strengthening our manuscript.

As a general clarification, we updated the station names for Nuup Kangerlua from GF6, GF7, etc., which referenced the Danish name Godthåbsfjord, to NK6, NK7, etc., to more clearly reflect their association with Nuup Kangerlua and enhance consistency throughout the manuscript.

We therefore added:

Line 137: It is important to note that earlier studies (e.g., Mortensen et al., 2011, 2014, 2018; Meire et al., 2015, 2017; Stuart-Lee et al., 2021, 2023) referred to the same stations in Nuup Kangerlua using the prefix "GF", derived from the Danish name "Godthåbsfjord". In this study, we use the prefix "NK" instead, to reflect the Greenlandic name "Nuup Kangerlua".

To avoid confusion, we updated this naming (NK) in your comments as well. We also numbered your questions to structure our answers.

**(1) In Fig. A1, it looks as if 210PB_xs was very low (and almost constant) for e.g. NK13 and NK6, how were sedimentation rates derived there?**

Following your comment, we chose to omit the sedimentation rate estimate of NK6, because (1) it is not a deep profile; only 10 cm deep and (2) $^{210}Pb_{xs}$ activity is indeed very low. This may be due to erosion conditions at the sill slope where older sediment material (characterized by low $^{210}Pb_{ex}$ activity) surfaces due to removal of younger sediment or older sediment has been transported to this location, which is also visible in NK7 and 9 profiles. The omission of the low MAR estimate at station NK6 resulted in a slightly higher average organic carbon burial rate for Nuup Kangerlua: $18.0 \pm 1.6$ g m$^{-2}$ yr$^{-1}$ versus the previously calculated $14.1 \pm 1.6$ g m$^{-2}$ yr$^{-1}$.

For station NK13 on the other hand, we opt to keep our estimate. MAR and SAR have been calculated the same way as for cores of stations AM10, NK12 and NK12 (see reply to question (3)). It is relatively close to the estimate of Pelikan et al. (2019) who sampled a sediment core at a distance of about 2.5 km west from our location, NK13. They estimated an age of ~200 years at the bottom of a 569 cm long sediment core, which comes down to an average sedimentation rate of ~2.85 cm yr$^{-1}$. Therefore, taking the lower SAR of 1.50 cm yr$^{-1}$ that we calculated would be more conservative. However, it should indeed be stressed in the manuscript that the MARs for stations NK13, NK12, NK10, AM10, which were obtained using the CF:CS calculation method, are less robust than the stations where the CSR method could be applied (as addressed in more depth in one of the answers below). Therefore we added: Line 366: "Note that the accumulation rates at stations NK10, NK12, NK13, and AM10 are estimates based on the CF:CS model and were not validated with independent time markers (Smith 2001; Barsanti 2020). Therefore, these estimates should be confirmed in future studies."

Moreover, we recalculated the MARs obtained by CRS dating as we first took the arithmetical average of the MARs. However, this approach did not account for the fact that, because MARs were not constant, each sediment layer corresponds to a different time span and should therefore be assigned a different statistical weight. The simplest way to give an average MAR (or SAR) in CRS dating, is to calculate the ratio depth/age in a deep layer. Therefore, we chose layers where the ages calculated using $^{210}Pb_{ex}$ method were confirmed by the $^{137}Cs$ profiles, hence the average value during approximately the last 60 years:

AM5:  A depth of $19 \pm 1$ cm or $5.45 \pm 0.25$ g cm$^{-1}$ (2021-1966) corresponds to $55 \pm 2$ years.

MAR=$1.0 \pm 0.1$ kg m$^{-2}$ yr$^{-1}$ & SAR = $3.5 \pm 0.2$ mm yr$^{-1}$

AM8:   A depth of 15 ±1 cm or 6.1± 0. 5 g cm$^{-1}$ (2022-1963) corresponds to 59 ± 3 years.

   MAR=1.1 ± 0.1 kg m$^{-2}$ yr$^{-1}$ & SAR = 2.6 ± 0.2 mm yr$^{-1}$

NK7:   A depth of 25 ± 1 cm or 14.53 ± 0. 55 g cm$^{-1}$ (2021-1960) corresponds to 61 ± 5 years.

   MAR=2.4 ± 0.2 kg m$^{-2}$ yr$^{-1}$ & SAR = 4.1 ± 0.4 mm yr$^{-1}$

NK9:   A depth of 29 ±1 cm, 19.14 ± 0. 85 g cm$^{-1}$ (2022-1961) corresponds to 61 ± 3 years.

   MAR=3.1 ± 0.2 kg m$^{-2}$ yr$^{-1}$ & SAR = 4.8 ± 0.3 mm yr$^{-1}$

An overview in table format is:

| Previous values | | | | Updated values (in line 369) | | | |
|---|---|---|---|---|---|---|---|
| Station | MAR (kg m$^{-2}$ yr$^{-1}$) | SAR (mm yr$^{-1}$) | OCBR (g m$^{-2}$ yr$^{-1}$) | Station | MAR (kg m$^{-2}$ yr$^{-1}$) | SAR (mm yr$^{-1}$) | OCBR (g m$^{-2}$ yr$^{-1}$) |
| GF13 | 14.1 ± 3.5 | 15.0 ± 3.7 | 27.5 ± 8.3 | NK13 | 14.1 ± 3.5 | 15.0 ± 3.7 | 27.5 ± 8.3 |
| GF12 | 5.9 ± 1.0 | 7.1 ± 1.2 | 9.6 ± 1.7 | NK12 | 5.9 ± 1.0 | 7.1 ± 1.2 | 9.6 ± 1.7 |
| GF10 | 7.0 ± 0.1 | 8.3 ± 1.1 | 29.4 ± 4.0 | NK10 | 7.0 ± 0.1 | 8.3 ± 1.1 | 29.4 ± 4.0 |
| GF9 | 3.4 ± 0.1 | 5.1 ± 1.4 | 19.1 ± 5.2 | NK9 | 3.1 ± 0.2 | 4.8 ± 0.3 | 17.5 ± 0.6 |
| GF7 | 2.6 ± 0.1 | 3.5 ± 1.3 | 6.5 ± 2.7 | NK7 | 2.4 ± 0.2 | 4.1 ± 0.4 | 5.9 ± 0.8 |
| GF6 | 1.8 ± 0.1 | 1.9 ± 0.8 | 1.5 ± 0.6 | AM10 | 4.0 ± 2.8 | 5.2 ± 2.0 | 9.9 ± 5.0 |
| AM10 | 4.0 ± 2.8 | 5.2 ± 2.0 | 13.1 ± 5.0 | AM8 | 1.1 ± 0.1 | 2.6 ± 0.2 | 17.7 ± 0.3 |
| AM8 | 1.1 ± 0.0 | 2.4 ± 0.0 | 17.2 ± 0.3 | AM5 | 1.0 ± 0.1 | 3.5 ± 0.2 | 21.0 ± 1.1 |
| AM5 | 1.1 ± 0.0 | 2.9 ± 0.3 | 23.9 ± 2.3 | | | | |

**(2)** The above outcome also addresses your following concern: "***Moreover, in Table 2, AM8 and AM5 have uncertainties of 0.0.***"

As a result, average OCBR for Nuup Kangerlua and Ameralik are:

| | Recalculated OCBR | Previous OCBR |
|---|---|---|
| Nuup Kangerlua | 18.0 ± 1.6 g m$^{-2}$ yr$^{-1}$ | 14.1 ± 1.6 g m$^{-2}$ yr$^{-1}$ |
| Ameralik | 16.2 ± 1.7 g m$^{-2}$ yr$^{-1}$ | 16.5 ± 1.7 g m$^{-2}$ yr$^{-1}$ |

Please note that the higher value obtained for Nuup Kangerlua is mainly due to, as discussed above, the exclusion of the low MAR of station NK6.

References are:

Barsanti, M., Garcia-Tenorio, R., Schirone, A., Rozmaric, M., Ruiz-Fernández, A. C., Sanchez-Cabeza, J. A., and Osvath, I.: Challenges and limitations of the [210]Pb sediment dating method: Results from an IAEA modelling interlaboratory comparison exercise, Quaternary Geochronology, 59, 101093, https://doi.org/10.1016/j.quageo.2020.101093, 2020.

Pelikan, C., Jaussi, M., Wasmund, K., Seidenkrantz, M., Pearce, C., Kuzyk, Z. Z. A., Herbold, C. W., Røy, H., Kjeldsen, K. U., and Loy, A.: Glacial runoff promotes deep burial of sulfur cycling-associated microorganisms in marine sediments, *Frontiers in Microbiology*, 10, 2558, https://doi.org/10.3389/fmicb.2019.02558, 2019.

Smith J (2001) Why should we believe 210Pb sediment geochronologies? J Environ Radioact 55: 121–123

(3) **"Log transformed $^{210}$Pbxs activities were plotted against the cumulative dry mass depth (g cm$^{-2}$) of the sediment per station." It would be good to show these plots.**

The following figures were added to the appendix to provide more clarity on the methodology:

[Figure]

**Figure A2.** The natural logarithm of the activity of $^{210}$Pb$_{ex}$ is plotted against the cumulative mass depth with the linear blue line representing CF:CS fitting for stations AM10, NK10, 12 and 13.

[Figure]

**Figure A3.** Age–depth models for sediment cores from stations AM5, AM8, NK7, and NK9, constructed using the Constant Rate of Supply (CRS) model based on $^{210}Pb_{ex}$ activity. Black circles represent modeled sediment ages plotted against cumulative mass depth (g cm$^{-2}$), with error bars showing ±1σ uncertainties. Red squares indicate the depth of the $^{137}Cs$ activity peak (1963), used as an independent chronological marker for model validation.

**(4) Also, why was the CRS model applied to some of the cores? For consistency (and if only an average sedimentation rate is required), for all cores the CF:CS model should be used.**

In a very dynamic environment such as fjords with mass discharge from glaciers, we suppose that the CRS method offers a higher chance to describe the sedimentation processes. This model gave good geochronology in the cores shown in figure A3, where every $^{210}Pb$ age is confirmed by $^{137}Cs$ measurements: the average sedimentation rate calculations in AM5, AM8, NK7 and NK9 are the most robust estimates, as an additional independent tracer ($^{137}Cs$) could be used.

Vice versa, the absence of a clear $^{137}Cs$ peak in the other stations (NK10, 12, 13 and AM10) did not allow to confirm the CRS dating and, since we needed an estimate of sedimentation rate, we applied the CF:CS model. It would be indeed better to emphasize that MAR and SAR values of stations NK10, NK12, NK13 and AM10 based on CF:CS dating are estimates that should be verified in future research. Therefore, we adapted the "2.4.1 Organic carbon burial rate" section in the Materials and Methods section:

Line 261: "Log-transformed $^{210}Pb_{ex}$ activities were plotted against cumulative dry mass depth (g cm$^{-2}$) for each station. Sedimentation rates at stations AM5, AM8, NK7, and NK9 were determined using the constant rate of supply (CRS) model (Appleby, 2001), as a distinct increase in $^{137}Cs$ was detected at these sites (Fig. A1), supporting the CRS-based chronology. The observed increase in $^{137}Cs$ activity is attributed to global fallout from atmospheric nuclear weapons testing, which peaked in 1963. In contrast, the CF:CS (constant flux:constant sedimentation) model (Sanchez-Cabeza and Ruiz-Fernández, 2012) was applied to stations NK10, NK12, NK13, and AM10, where the $^{210}Pb_{ex}$ profiles exhibited approximately exponential trends but lacked a clearly defined $^{137}Cs$ peak. For these stations, log-transformed $^{210}Pb_{ex}$ activities were plotted against cumulative dry mass depth (g cm$^{-2}$) for each station.  As a result, the sedimentation rate estimates for these stations should be treated with caution and verified in future studies. Mass accumulation rates (MAR, kg m$^{-2}$ yr$^{-1}$) were derived from the slope of the linear regression (for CF:CS) or from the CRS model output. Bulk sediment accumulation rates (SAR, mm yr$^{-1}$) were calculated by

dividing MAR by the average bulk density at each station. Organic carbon burial rates (OCBRs) were then calculated by multiplying MAR by the TOC content at the 9 – 10 cm sediment layer."

**(5) Furthermore, how was the lack of bioturbation determined? From the Cs profiles in Fig. A1, it does look as if mixing may have affected at least some of the cores.**

We stated "No bioturbated or mixed upper layer was observed in the profiles." This indeed required a more elaborated explanation and was changed to (lines 281-288): "We did not apply corrections for bioturbation or mixing processes, as the $^{210}Pb_{ex}$ profiles do not show evidence of such activity in the upper sediment layers. However, these processes cannot be conclusively ruled out, particularly since some of the $^{137}Cs$ profiles feature broad activity peaks. Nonetheless, the 210Pb-derived chronology appears to be supported by the $^{137}Cs$ profiles in AM5, AM8, NK7 and NK9 (Smith, 2001; Barsanti et al., 2020). The broad $^{137}Cs$ curves or inflections, marking sustained elevation in $^{137}Cs$ activity after an initial increase followed by a gradual decrease over time, are therefore more likely explained by continued exposure of settling particles to residual $^{137}Cs$ in the overlying water after 1963. As a result, younger sediment layers also contain measurable amounts of $^{137}Cs$, smearing the signal across multiple horizons. This phenomenon has been observed in other marine settings (Tamburrino et al. 2019) and even in lake sediments (Drexler et al., 2018).

Following references were added to the reference list:

Barsanti, M., Garcia-Tenorio, R., Schirone, A., Rozmaric, M., Ruiz-Fernández, A. C., Sanchez-Cabeza, J. A., and Osvath, I.: Challenges and limitations of the $^{210}Pb$ sediment dating method: Results from an IAEA modelling interlaboratory comparison exercise, Quaternary Geochronology, 59, 101093, https://doi.org/10.1016/j.quageo.2020.101093, 2020.

Drexler, J. Z., Fuller, C. C., and Archfield, S.: The approaching obsolescence of $^{137}Cs$ dating of wetland soils in North America, Quaternary Science Reviews, 199, 83–96, https://doi.org/10.1016/j.quascirev.2018.08.028, 2018.

Smith, J. N.: Why should we believe $^{210}Pb$ sediment geochronologies?, Journal of Environmental Radioactivity, 55, 121–123, https://doi.org/10.1016/S0265-931X(01)00110-2, 2001.

Tamburrino, S., Passaro, S., Barsanti, M., Schirone, A., Delbono, I., Conte, F., Delfanti, R., Bonsignore, M., Del Core, M., Gherardi, S., en Sprovieri, M.: Pathways of inorganic and organic contaminants from land to deep sea: The case study of the Gulf of Cagliari (W Tyrrhenian Sea), The Science Of The Total Environment, 647, 334–341, https://doi.org/10.1016/j.scitotenv.2018.07.467, 2019.

**(6) Another, smaller point: the authors state that their findings regarding higher OC content in the LTG fjord came as a surprise – yet comparisons with literature data (Figure 5) reveal that this is not so different from earlier studies in other regions.**

Taking into account your remark, we rephrased lines 391-398 within the discussion: "So far, studies comparing MTG and LTG fjord systems are limited (Koziorowska et al., 2015; Laufer-Meiser et al., 2021). These studies suggest that MTG fjords tend to exhibit higher OC accumulation, as indicated by elevated OC content in surface sediments. However, when comparing the LTG system Ameralik and the MTG system Nuup Kangerlua with datasets from other fjords (Smith et al., 2002; Thamdrup et al., 2007; Koziorowska et al., 2015; Cui et al., 2016; Faust and Knies, 2019; Włodarska-Kowalczuk et al., 2019; Laufer-Meiser et al., 2021), we observed that surface sediment OC content in LTG and even non-glaciated fjords can be comparable to that of MTG systems across the (sub-)Arctic region (Fig. 5A).

Nevertheless, it is important to note that LTG fjords are underrepresented in current datasets, and low-glacial-activity MTG systems may bias comparative interpretations."

**(7) Abstract: The last two sentences could be combined as they are a bit redundant at the moment. The available space could then be allocated to mention e.g. results for OC composition or provide more detail (orders of magnitude) for sedimentation rates and/or OCBR, or some info on which parameters were measured for this study.**

Good point. We applied your comment in:

Line 17: This study compares sediment TOC, TN, and Chl-a content as well as $\delta^{13}C$, $\delta^{15}N$ and organic carbon burial rates (OCBRs) in two neighbouring Greenland fjords, Nuup Kangerlua, influenced by marine-terminating glaciers (MTGs), and Ameralik, dominated by land-terminating glaciers (LTGs), to explore the effects of both types of glaciers on sediment carbon dynamics.

Line 24: Despite different glacial regimes, the two investigated fjord systems showed similar traits with OC predominantly of marine origin and similar OCBRs of $18.0 \pm 1.6$ and $16.2 \pm 1.7$ g m$^{-2}$ yr$^{-1}$ in Nuup Kangerlua and Ameralik, respectively. Higher Chl-a and OC contents were recorded in the sediments of outer and mid Ameralik compared to those in Nuup Kangerlua. The results underscore that benthic – pelagic coupling in glacial fjords is complex, emphasizing the need for further research to disentangle the interactions driving primary production, lateral and vertical OC transport, as well as OC degradation and preservation in fjord sediments.

**(8) Figure 1: Sample IDs could be shown on the map.**

We improved Figure 1 (line 122) based on your comment and the other two reviewers by:

- Adding a legend, depicting stations sampled in Nuup Kangerlua and stations in Ameralik in a different color.
- Adding station labels and adjusting the symbol of Nuuk to differentiate it from the station symbols.
- Situating the outer, mid and inner zone in the (b) panel.
- Enlarging (a) and (b) panel.

[Figure]

**Figure 1. (a)** Map showing sampling locations in Nuup Kangerlua (fed by three marine-terminating glaciers and three land-terminating glaciers) and Ameralik (receiving meltwater from a land-terminating glacier). Greenland Ice Sheet (GrIS) is depicted in white. **(b)** Water depth profiles along-axis (white dashed lines) Nuup Kangerlua (top) and Ameralik (bottom). Both fjord basins are divided in an outer, mid and inner section behind the entrance sill(s).

**(9) Figure 4: A second panel with C:N vs d13C could be added, as C:N ratios have also been commonly employed to distinguish terrestrial from marine OC.**

Thank you for this insightful suggestion. We agree that plotting $\delta^{13}C$ against C:N ratios adds value, as both are commonly used to distinguish between terrestrial and marine organic carbon sources. We have therefore revised Figure 4 (line 358) including other remarks of other reviewers. We improved this figure by:

- Enlarging all elements of the figure.
- Adding a second panel, displaying $\delta^{13}C$ against C:N ratio. We indicated the marine and terrestrial end-member $\delta^{13}C$ values used in this study to determine the proportion of marine and terrestrial OC.
- Colors referring to Nuup Kangerlua and Ameralik were adapted to follow the color code of the map (Figure 1; line 122).
- In panel (a), we extended the marine and terrestrial ranges from $\delta^{13}C$ -23 ‰ to $\delta^{13}C$ -24‰ and from $\delta^{13}C$ -25 ‰ to $\delta^{13}C$ -21‰ (following Lamb et al., 2006), respectively.

[Figure]

**Figure 4. (a)** $\delta^{13}C$ (‰ deviations from V-PDB) values plotted against $\delta^{15}N$ (‰ deviations from air) values of the POM present in the sediment for the different station of Ameralik (filled symbols) and Nuup Kangerlua (open symbols). Typical marine and terrestrial ranges of $\delta^{13}C$ (lamb et al., 2006) and $\delta^{15}N$ (Zaborska et al., 2018) are indicated with rectangles. **(b)** $\delta^{13}C$ plotted against C:N ratios. Ranges of marine and freshwater POC, and C3 terrestrial plants are displayed as rectangles for reference (values taken from Lamb et al., 2006). Marine (M) and terrestrial (T) $\delta^{13}C$ end-members used in this study are indicated with arrows.

In addition we added to the text, section "2.3.1 Calculation of marine organic carbon fraction": lines 212 – 219: Stable isotope composition in addition to C:N ratios of settled organic matter in fjord sediments has been used in multiple studies to estimate the proportion of marine versus terrestrially derived organic matter (St-Onge and Hillaire-Marcel, 2001; Koziorowska et al., 2015; Smeaton & Austin, 2017; Zaborska et al., 2018; Faust and Knies, 2019; Limoges et al., 2020; Placitu et al., 2024). Terrestrial organic matter, primarily derived from vascular plants, tends to have higher C:N ratios (> 12) and more depleted $\delta^{13}C$ values (-25 to -30 ‰ $\delta^{13}C$) due to the dominance of lignin-rich, cellulose-based material and the use of C3 photosynthesis pathways (Lamb et al., 2006). In contrast, marine organic matter, originating from phytoplankton and other aquatic organisms, typically shows lower C:N ratios and less negative $\delta^{13}C$ values (-18 to -24 ‰ $\delta^{13}C$), reflecting a protein-rich composition and different carbon fixation mechanisms (Lamb et al., 2006).

**(10)** **L161-170: For the proportions of terrestrial vs marine OC, the selection of endmember values need more explanation: "These end-members were derived from Northern and Mid-Norway fjord sediments" – how? Also, those values should have some uncertainties, which would then propagate to the fraction estimates. In Figure 4, the endmembers appear to have uncertainties.**

We added the following to the "2.3.1 Calculation of marine organic carbon fraction section": Lines 227-234: The catchments of both fjords consist of tundra shrub vegetation, which are typically $C_3$ plants. Published $\delta^{13}C$ values for terrestrial plant material in Greenland remain limited, but available data indicate a range of -33.9‰ to -26.9‰ (Thompson et al., 2018). However, due to the scarcity of $\delta^{13}C$ records specific to Greenland's marine organic matter, terrestrial vegetation and soil, we adopted end-member values from nearby Arctic and sub-Arctic systems. For the marine end-member, we used a $\delta^{13}C$ value of −20.6‰, consistent with those reported in Svalbard studies by Winkelman and Knies, and Koziorowska et al. (2015). We used the marine end-member value from Faust and Knies (2019), originally applied in sub-Arctic Norwegian fjords, as it falls within the broader $\delta^{13}C$ range of Arctic terrestrial organic matter (-35‰ to -25‰) reported by Kuliński et al. (2014).

The data derived from these publications did not contain uncertainty estimates, but we used the reported values to indicate the range of end-member values (dashed zones in figure 4).

We added following references:

Limoges, A., Weckström, K., Ribeiro, S., Georgiadis, E., Hansen, K. E., Martinez, P., Seidenkrantz, M., Giraudeau, J., Crosta, X., en Massé, G.: Learning from the past: Impact of the Arctic Oscillation on sea ice and marine productivity off northwest Greenland over the last 9,000 years, Global Change Biology, 26, 6767–6786, https://doi.org/10.1111/gcb.15334, 2020.

Thompson, H. A., White, J. R., and Pratt, L. M.: Spatial variation in stable isotopic composition of organic matter of macrophytes and sediments from a small Arctic lake in west Greenland, Arctic Antarctic And Alpine Research, 50, https://doi.org/10.1080/15230430.2017.1420282, 2018.

St-Onge, G. and Hillaire-Marcel, C.: Isotopic constraints of sedimentary inputs and organic carbon burial rates in the Saguenay Fjord, Quebec, Marine Geology, 176, 1–22, https://doi.org/10.1016/s0025-3227(01)00150-5, 2001.

**Technical corrections:**

**(11)** **Short summary: "necessarily" instead of "necessary"**

Adjusted in the text.

**(12)** **Abstract: L21, 23 & 25 "organic carbon" should be "OC"**

Adjusted in the text.

**(13)** **Figure A2: In the caption, it reads "Orange and black colors represent end of summer 2021 and spring 2022", yet there is no orange in the figure.**

Figure caption changed to: "… Grey and black colors represent end of summer 2021 and spring 2022, respectively. Data from the two seasons is available for stations NK12, NK7, AM10 and AM5." (Line 665).

Based on comments of other two reviewers, we reformulated large parts of the discussion and we reorganized it according to the following structure:

- **4.1 Surface sediment OC content** with a sub-section on **OC origin** (4.2.1), putting more emphasis on what can be derived from our data.
- **4.2 Organic carbon burial rates**, which was set as the 2nd sub section instead of the last paragraph. We also included primary production values from literature.
- We assembled our 3 main hypotheses in sub-sections within **4.3 Pelagic and geomorphological influence on OC burial:**
    - o **4.3.1 OC preservation conditions**, which was condensed and more focused on the bottom water temperature differences we measured between the fjords. We removed the hypothesis regarding a difference in OC preservation related to Fe and Mn in both fjords from the discussion due to limited data to support this hypothesis.
    - o We condensed section **4.3.2 Transport dynamics.** This section was included to provide a more nuanced view of MTG-induced upwelling, acknowledging that nutrient enrichment may also originate from upwelling at the fjord mouth, potentially influencing the patterns observed in our study
    - o We condensed the **4.3.3 Food web OC uptake** section.
- We incorporated the suggestion of Anonymous Referee #1 to add a section on Recommendations for future research.

To avoid an overly long rebuttal letter, we refer the reviewer to our rewritten discussion in the manuscript. (From line 371 onwards).

---

## Author Comment (AC2)

We are grateful to you for the time you invested in providing detailed and helpful feedback. Your comments have been invaluable in refining and strengthening our manuscript.

As a general clarification, we updated the station names for Nuup Kangerlua from GF6, GF7, etc., which referenced the Danish name Godthåbsfjord, to NK6, NK7, etc., to more clearly reflect their association with Nuup Kangerlua and enhance consistency throughout the manuscript.

We therefore added:

Line 137: It is important to note that earlier studies (e.g., Mortensen et al., 2011, 2014, 2018; Meire et al., 2015, 2017; Stuart-Lee et al., 2021, 2023) referred to the same stations in Nuup Kangerlua using the prefix "GF", derived from the Danish name *Godthåbsfjord*. In this study, we use the prefix "NK" instead, to reflect the Greenlandic name *Nuup Kangerlua*.

To avoid confusion, we updated this naming (NK) in your comments as well. We also numbered your questions and highlighted them in bold to clearly structure our responses.

**Abstract**

**(1) L1 – Fjords play a crucial role in the burial of OC – not they are crucial – same as L26 of introduction.**

We applied your remark. In addition, agreeing with your comment on Blue carbon (as you commented in the introduction: comment (3)) and based on the paper you recommended (Howard et al., 2023), we removed "blue carbon" as it is indeed not the most correct use given the link with policy and management/mitigation practices, which is not the focus of this paper.

Line 14: Fjord systems play a crucial role in the burial and long-term storage of organic carbon (OC).

**(2) L23-25 – duplicated sentences – could be combined.**

Taking your comment and the comment of another reviewer into account, the last part of the abstract was rewritten as follows:

Lines 24-29: Despite different glacial regimes, the two investigated fjord systems showed similar traits with OC predominantly of marine origin and similar OCBRs of $18.0 \pm 1.6$ and $16.2 \pm 1.7$ g m-2 yr-1 in Nuup Kangerlua and Ameralik, respectively. Higher Chl-a and OC contents were recorded in the sediments of outer and mid Ameralik compared to those in Nuup Kangerlua. The results underscore that benthic – pelagic coupling in glacial fjords is complex, emphasizing the need for further research to disentangle the interactions driving primary production, lateral and vertical OC transport, as well as OC degradation and preservation.

**Introduction**

**(3) L28 'Blue carbon' – this isn't defined but should be. Sediments are currently understood to be 'emerging' blue carbon systems and are not accepted as 'actionable' blue carbon systems (See Howard et al., 2023)? The term isn't mentioned again so why use it at all? Regarding the reference to Smith et al., 2015: Does the annual blue carbon burial include mangroves, seagrass and saltmarsh? Smith's paper discussed fjords in the context of global sedimentary carbon only.**

As earlier mentioned in our reply to comment (1), we replaced "blue carbon" by "carbon":

Line 35: Fjord systems play a crucial role in burial and long-term storage of organic carbon (OC), contributing to approximately a tenth of the annual carbon burial (Smith et al., 2015).

**(4) L32 – could also mention southern-hemisphere countries that have looked at OC burial in fjords -e.g. Chile.**

We added research on OC burial in the Southern hemisphere:

Line 40: In the Southern Hemisphere, estimates of OC burial in fjord systems are comparatively sparse and largely confined to a few regions, including Patagonia (Sepúlveda et al., 2011), South Georgia (Berg et al., 2021), Antarctica (Eidam et al., 2019), and New Zealand (Hinojosa et al., 2014; Cui et al., 2016b).

And adjusted the subsequent sentence (line 43) to enhance connection between sentences:

Despite this growing body of research in both hemispheres, Greenland remains markedly underrepresented in global carbon budgets (Smith et al., 2015), even though its coast is fringed by a myriad of fjords, including some of the most extensive fjord systems in the Arctic.

Added references are:

Berg, S., Jivcov, S., Kusch, S., Kuhn, G., White, D., Bohrmann, G., Melles, M., en Rethemeyer, J.: Increased petrogenic and biospheric organic carbon burial in sub-Antarctic fjord sediments in response to recent glacier retreat, Limnology And Oceanography, 66, 4347–4362, https://doi.org/10.1002/lno.11965, 2021.

Eidam, E. F., Nittrouer, C. A., Lundesgaard, Ø., Homolka, K. K., en Smith, C. R.: Variability of Sediment Accumulation Rates in an Antarctic Fjord, Geophysical Research Letters, 46, 13271–13280, https://doi.org/10.1029/2019gl084499, 2019.

Sepúlveda, J., Pantoja, S., & Hughen, K. A. (2010). Sources and distribution of organic matter in northern Patagonia fjords, Chile (~44–47°S): A multi-tracer approach for carbon cycling assessment. *Continental Shelf Research*, *31*(3–4), 315–329. https://doi.org/10.1016/j.csr.2010.05.013

**(5) L57 – depends on terrestrial vegetation found within the catchment – rather than growth specifically?**

We agree with your remark and changed "terrestrial vegetation growth" to "terrestrial vegetation": Line 69 "In general, the effectiveness of trapping OC varies among fjords and depends on (1) the productivity of the fjord waters, particularly phytoplankton growth, as well as terrestrial vegetation in the catchment, …"

**(6) L60 – 62 – sediment type also important. The type and spatial distribution of sediment will be somewhat controlled by fjord geomorphology and current dynamics already mentioned – but %clay and presence of Fe minerals within the sediment has a control on the availability of OC and its degradation potential (see: Lalonde et al., 2012; Hunt et al., 2020; Moore et al., 2023)**

Thank you for your insight. We extended line 74: (3) factors limiting the degradation of settled OC, among which the refractory nature of OC (Koziorowska et al., 2015; Zaborska et al., 2018), adsorption to clay minerals (Kennedy and Wagner, 2011; Daidu, 2006), binding with reactive iron and manganese phases (Lalonde et al., 2012; Faust et al., 2021; 2023; Moore et al., 2023), sedimentation rate (Placitu et al., 2024; Watts et al., 2024) and bottom water redox conditions (Hinojosa et al., 2014).

**(7) Line 66 – sequestration refers to the removal of carbon dioxide from the atmosphere. Within sediments, as passive receivers of organic matter, the term OC accretion of burial is more appropriate.**

Thank you for the clarification. We agree that "burial" or "accretion" is more accurate in this context, as sediments indeed act as passive receivers of organic carbon. We have revised the text accordingly at line 79: "…may lead to increased carbon burial in fjord sediments compared to LTG-influenced fjord systems…"

**Materials and Methods**

**(8) Figure 1 – could the red dots instead be changed to one colour to represent the LTGs and another colour to represent the MTGs? Station labels would be helpful**

We improved Figure 1 (line 122) by:

- Adding a legend, depicting stations sampled in Nuup Kangerlua and stations in Ameralik in a different color.
- Adding station labels and adjusting the symbol of Nuuk to differentiate it from the station symbols.
- Situating the outer, mid and inner zone in the (b) panel.
- Enlarging (a) and (b) panel.

[Figure]

[Figure]

**Figure 1. (a)** Map showing sampling locations in Nuup Kangerlua (fed by three marine-terminating glaciers and three land-terminating glaciers) and Ameralik (receiving meltwater from a land-terminating glacier). Greenland Ice Sheet (GrIS) is depicted in white. **(b)** Water depth profiles along-axis (white dashed lines) Nuup Kangerlua (top) and Ameralik (bottom). Both fjord basins are divided in an outer, mid and inner section behind the entrance sill(s).

**(9) Section 2.2.1 – more appropriate to call this 'Core Sampling'?**

Section 2.2.1 was changed to "Core sampling and processing" (line 141).

**(10)       L127 - Why was the 2021 campaign more exploratory?**

The first campaign had fewer stations since it was the first that sediments were sampled in these fjords, hence the exploratory character of the campaign.

However, we adjusted line 145: Fewer sediment stations were sampled in 2021 compared to 2022; however, bottom water temperature measurements were obtained in both years (Table 1).

**(11)       2.3 - More detail needed here. Think about how someone would replicate your analysis exactly. E.g., not sure what has been done here: 'Porosity was obtained by dividing the porewater volume by the wet sediment volume.'**

Implementing your comment together with the comments of the other 2 reviewers, the 2.3 Sediment analysis section was rewritten as follows: line 159-210:

Grain size, porosity and dry bulk density were measured to provide insights into the physical structure and depositional environment of the sediment column. High porosity typically indicates fine-grained, loosely packed sediments with higher water content, which are common in low-energy depositional environments. Conversely, lower porosity may suggest coarser, more compacted sediments, potentially reflecting higher-energy conditions or post-depositional consolidation. Grain size distribution was determined on oven-dried samples (at 60 °C for 48 h). After homogenization, coarse material > 2 mm was

removed by sieving. A subsample of 0.1 – 1 g was resuspended in water and analyzed using a Malvern Mastersizer 2000 with the Hydro 2000S module (size range: 0.02 – 2000 µm), which operates based on laser diffraction. The sample is sonicated for 60 s to prevent flocculation of clays before it goes through the laser prior measurement. No pretreatment to remove organic or inorganic carbon was performed prior to analysis. Grain size fractions were classified according to the Wentworth scale (1922) as clay (< 4 µm), silt (4 – 63 µm), and sand (63 – 500 µm).

Results were categorized in clay (<4 mm), silt (4–63 mm), and sand (63–500 mm) fractions conform Wentworth scale classification (1922). Sediment porosity was estimated gravimetrically using a modified water displacement method. A pre-weighed 10 mL graduated measuring cylinder was filled with 2 – 5 g of homogenized wet sediment. After careful addition of ultrapure water (Milli-Q) to the 10 mL mark (measured by the lower meniscus), the cylinder was weighed again. The sample was then dried at 80 °C for ~48 h and reweighed. Porosity ($\phi$) was calculated based on the difference between the wet sediment weight and the dry sediment weight (i.e., the mass of porewater), recalculated to pore water volume through correction for salinity and divided by the estimated volume occupied by the sediment (calculated as 10 ml minus the volume of water added). Dry bulk density was obtained by dividing dry mass by bulk volume.

To determine total carbon (TC), total organic carbon (TOC), and total nitrogen (TN), sediment samples were oven-dried at 60 °C for 48 hours ground using mortar and pestle, and homogenized. Between 20 – 35 mg of dried and homogenized sediment was weighed and placed into pre-weighed silver cups. For TC and TN analysis, silver cups were sealed by folding with tweezers into compact spheres to ensure complete combustion. For TOC analysis, carbonate removal was performed by stepwise acidification: 2 – 3 drops of increasing concentrations of HCl (1%, 2%, 5%, and 10%) were added sequentially using a glass pipette. After each addition, samples were dried at 60 °C for 1 to 2 hours. This procedure was repeated over 2 – 3 days until no bubbling was observed and carbonate removal was confirmed. All measurements were conducted using a Flash 2000 NC Sediment Analyzer (Interscience), which quantifies carbon and nitrogen via dynamic flash combustion and chromatographic separation. From these data, the molar C:N ratios were calculated dividing TOC by TN and inorganic carbon (IC) was determined by subtracting TOC from TC. To investigate the origin of the organic matter (see 2.3.1), stable isotope composition ($\delta$13C and $\delta$15N) was measured with an elemental analyzer (Thermo Flash EA1112 elementanalyzer) coupled to an isotope ratio mass spectrometer (Thermo Finnigan Delta V, IRMS). Prior to analysis, the same steps were followed as for TOC and TN analysis, except samples were freeze dried.

Samples were dried and homogenized, then analyzed for total sedimentary carbon (TC) and total nitrogen (TN). After decalcification with 37 % HCl, total organic carbon (TOC) was also measured. All measurements were conducted using a Flash 2000 NC Sediment Analyzer (Interscience). From these data, the molar CN ratios were calculated and inorganic carbon (IC) was determined by subtracting TOC from TC. To investigate the origin of the organic matter (see further), stable isotope composition ($\delta$13C and $\delta$15N) was measured with an elemental analyzer (Thermo Flash EA1112 elementanalyzer) coupled to an isotope ratio mass spectrometer (Thermo Finnigan Delta V, IRMS). To explore how glacier type affects marine primary productivity and whether and how it is incorporated in the sediment, we additionally measured, for each sediment slice, the content of chlorophyll-a (Chl-a) and of its degradation products (pheophorbide-a, and pheophytin-a, pheophorbide-a like, and pheophytin-a like following Wright and Jeffrey (1997). For pigments extraction, 2 mL acetone (90%) was added to 0.5 g freeze dried sediment under red light conditions preventing pigment degradation. The samples were subsequently sonicated for 30 s and incubated overnight at 4 °C in the dark to aid pigment release. Afterwards, the samples were centrifuged (10 min, 4 000 rpm, 4 °C) and the supernatant was passed through 0.2 µm PTFE filters.

Pigment separation was performed using an HPLC system (Agilent 1200 Infinity II, Agilent Technologies) equipped with a cooled auto-sampler, column oven, photodiode array detector, and fluorescence detector, following the method of Van Heukelem and Thomas (2001). Chlorophyll a and its degradation products were identified at 665 nm wavelength. Individual pigment concentrations were determined using the response factors of the respective standards. These different pigments were determined by the response factor of standard pigments as described by Van Heukelem and Thomas (2001). The ratio of Chl-a to Chloroplastic Pigment Equivalent (CPE, comprising the sum of all aforementioned pigments) was used as a proxy for the "freshness" or lability of photosynthetically produced organic matter (Schubert et al., 2005; Koho et al., 2008).

**(12) L155 – some more description as to why these parameters help to differentiate between terrestrial and marine-derived material would be helpful.**

We added the following to the 2.3.1 Calculation of marine organic carbon fraction section: line 212-219: Stable isotope composition in addition to C:N ratios of settled organic matter in fjord sediments has been used in multiple studies to estimate the proportion of marine versus terrestrially derived organic matter (St-Onge and Hillaire-Marcel, 2001; Koziorowska et al., 2015; Smeaton & Austin, 2017; Zaborska et al., 2018; Faust and Knies, 2019; Limoges et al., 2020; Placitu et al., 2024). Terrestrial organic matter, primarily derived from vascular plants, tends to have higher C:N ratios (> 12) and more depleted $\delta^{13}C$ values (-25 to -30 ‰ $\delta^{13}C$) due to the dominance of lignin-rich, cellulose-based material and the use of $C_3$ photosynthesis pathways (Lamb et al., 2006). In contrast, marine organic matter, originating from phytoplankton and other aquatic organisms, typically shows lower C:N ratios and less negative $\delta^{13}C$ values (-18 to -24 ‰ $\delta^{13}C$), reflecting a protein-rich composition and different carbon fixation mechanisms (Lamb et al., 2006).

**(13) L166 – Why were the top 2 cm of sediment only used?**

We took the upper 2 cm to be able to compare with literature as well as to avoid potential bias due to degradation processes. We added line 239: "…where $\delta^{13}C_i$ represents the surface sediment values (0 – 2 cm) of $\delta^{13}C_{org}$ of each sample, $\delta^{13}C_M$ is the marine end-member and $\delta^{13}C_T$ is the terrestrial end-member. Only the upper 0 – 2 cm was used to be able to compare with literature.

**(14) L168 – is the same terrestrial end member as Faust's appropriate here? What are the dominant vegetation type within these study fjords?**

We added the following to the "2.3.1 Calculation of marine organic carbon fraction section": Line 227-234: The catchments of both fjords consist of tundra shrub vegetation, which are typically $C_3$ plants. Published $\delta^{13}C$ values for terrestrial plant material in Greenland remain limited, but available data indicate a range of -33.9‰ to -26.9‰ (Thompson et al., 2018). However, due to the scarcity of $\delta^{13}C$ records specific to Greenland's marine organic matter, terrestrial vegetation and soil, we adopted end-member values from nearby Arctic and sub-Arctic systems. For the marine end-member, we used a $\delta^{13}C$ value of −20.6‰, consistent with those reported in Svalbard studies by Winkelman and Knies, and Koziorowska et al. (2015). We used the marine end-member value from Faust and Knies (2019), originally applied in sub-Arctic Norwegian fjords, as it falls within the broader $\delta^{13}C$ range of Arctic terrestrial organic matter (-35‰ to -25‰) reported by Kuliński et al. (2014).

We added following references:

Limoges, A., Weckström, K., Ribeiro, S., Georgiadis, E., Hansen, K. E., Martinez, P., Seidenkrantz, M., Giraudeau, J., Crosta, X., en Massé, G.: Learning from the past: Impact of the Arctic Oscillation on sea

ice and marine productivity off northwest Greenland over the last 9,000 years, Global Change Biology, 26, 6767–6786, https://doi.org/10.1111/gcb.15334, 2020.

Thompson, H. A., White, J. R., and Pratt, L. M.: Spatial variation in stable isotopic composition of organic matter of macrophytes and sediments from a small Arctic lake in west Greenland, Arctic Antarctic And Alpine Research, 50, https://doi.org/10.1080/15230430.2017.1420282, 2018.

St-Onge, G. and Hillaire-Marcel, C.: Isotopic constraints of sedimentary inputs and organic carbon burial rates in the Saguenay Fjord, Quebec, Marine Geology, 176, 1–22, https://doi.org/10.1016/s0025-3227(01)00150-5, 2001.

**(15)    L197 – Expand on how the 10 cm of sediment column was averaged.**

We adjusted line 291 to specify the mean: We examined differences between the two fjords and among stations in terms of sedimentary TOC and TN content, C:N ratio, Chl-a content, and the Chl-a:CPE ratio, using data from both the upper 2 cm of the sediment surface and the arithmetic mean of the upper 10 cm sediment column.

**(16)    L199 – How did you know the seasonal difference was insignificant?**

We clarified this in the text, in line 293: Data from summer (2021) and spring (2022) were combined and treated as replicates, as the difference between the two seasons was insignificant (Welch's ANOVA, $p >$ 0.05; see further). As a consequence, stations NK12, NK10, NK7, AM10 and AM5 have six replicates since they were sampled 200 in both seasons, while the other stations have three replicates as those stations were only sampled during spring 2022 (Table 1). Statistical analyses were performed using one-way ANOVA. Welch's ANOVA was applied when variances were unequal, and the Kruskal-Wallis test was used when normality assumptions were violated.

**Results**

**(17)    L210-215 – I don't agree that there is much difference between the sediment grain between NK13, NK12, NK10 and NK6 (inner and outer).**

We adjusted the text by nuancing the difference in grain size: line 306: In Nuup Kangerlua, the median grain size ($d_{0.5}$) exhibits a modest spatial trend from the inner to the outer fjord (Fig. 2). At the inner stations (NK13 and NK12), grain size remains relatively small (< 20 μm) and consistent with depth, reflecting a stable depositional environment dominated by fine particles. Grain size at the mid-fjord stations (NK10 and NK9) is slightly larger but still within the fine-silt range, indicating only subtle hydrodynamic variation. At the outer stations (NK7 and NK6), grain size increases slightly further and shows more variability with depth, which may reflect localized influence of bottom currents or episodic input of coarser particles near the fjord mouth. Overall, differences in grain size between stations are relatively small, but a general trend toward coarser material at the fjord's outer reaches is observable.

[Figure]

**Figure 2.** Sediment profiles of median grain size (μm), porosity and dry density (g cm-3) of Nuup Kangerlua (GFNK stations) and Ameralik (AM stations). Error bars represent SD (n = 3 NK6, NK9, NK13 and AM8 and n = 6 for NK7, NK10, NK12, AM5 and AM10). Only one replicate for porosity and dry density.

**(18)** **L220 – Would be helpful to expand upon why porosity and dry bulk density (not mentioned in the methods) have been measured and what they tell us about the sediment dynamics?**

The following was added to 2.3 Sediment analysis section: lines 159-162: Grain size, porosity and dry bulk density were measured to provide insights into the physical structure and depositional environment of the sediment column. High porosity typically indicates fine-grained, loosely packed sediments with higher water content, which are common in low-energy depositional environments. Conversely, lower porosity may suggest coarser, more compacted sediments, potentially reflecting higher-energy conditions or post-depositional consolidation.

The following was added after the description of porosity analysis: line 177: Dry bulk density was obtained by dividing dry mass by bulk volume.

In addition, dry bulk density was also needed to calculate MAR as it was plotted against cumulative dry mass depth as well as to obtain SAR as specified in line 270: Bulk sediment accumulation rates (SAR, mm yr⁻¹) were calculated by dividing MAR by the average bulk density at each station.

**Discussion**

**(19)      L277 – introduces the term lability for the first time. Can this be expanded upon somewhere – what does it mean, what parameters have measured it and why does it matter?**

We added the following to 2.3. Sediment analysis section: line 208: The ratio of Chl-a to Chloroplastic Pigment Equivalent (CPE, comprising the sum of all aforementioned pigments) was used as a proxy for the "freshness" or lability of photosynthetically produced organic matter (Schubert et al., 2005; Koho et al., 2008).

Koho, K. A., García, R., De Stigter, H. C., Epping, E., Koning, E., Kouwenhoven, T. J., en Van Der Zwaan, G. J.: Sedimentary labile organic carbon and pore water redox control on species distribution of benthic foraminifera: A case study from Lisbon–Setúbal Canyon (southern Portugal), Progress in Oceanography, 79, 55–82, https://doi.org/10.1016/j.pocean.2008.07.004, 2008.

Schubert, C. J., Niggemann, J., Klockgether, G., en Ferdelman, T. G.: Chlorin Index: A new parameter for organic matter freshness in sediments, Geochemistry Geophysics Geosystems, 6, https://doi.org/10.1029/2004gc000837, 2005.

**(20)      L320 – the temperature difference between the two fjord systems is a tangible difference and could be better highlighted. The discussion about Fe and Mn is interesting, but currently limited data to tease the two fjords apart.**

Following your comment, we removed the hypothesis of a difference in OC preservation related to Fe and Mn in both fjords as there is indeed too little data to formulate arguments. In addition, we rephrased the paragraph in the discussion on bottom water temperature differences, applying our measurements. Lines 445-458: "Since most of the OC deposited in both fjords is of marine origin, any differences in organic matter preservation between them are likely driven by environmental conditions rather than by differences in the nature of the organic material itself. The distinct geomorphology of Ameralik and Nuup Kangerlua, particularly their differing sill depths, likely shapes bottom water temperatures and may influence organic matter preservation within the fjords. Both fjords have no anoxic deep water masses and bottom water renewal occurs every one to two years (Mortensen, 2011; Stuart-Lee et al., 2021), but bottom water temperature differs. Ameralik's shallower sill depth (∼110 m) compared to Nuup Kangerlua (∼200 m) restricts the inflow of warmer, saltier coastal waters (Stuart-Lee et al., 2021). Consequently, during field sampling, bottom water temperatures in Nuup Kangerlua were consistently warmer than in Ameralik, particularly in spring, with average values of 1.33 °C and 0.53 °C, respectively. The lower bottom water temperatures in Ameralik may cause the observed higher pigment and OC preservation in AM5 by reducing microbial degradation and slowing remineralization processes. A comparative study of several Svalbard fjords suggested that relatively higher pigment content in sediments may be linked to lower bottom water temperatures (Krajewska et al., 2020). However, this hypothesis warrants further investigation, as Arctic microbial communities are well adapted to low temperatures, and mineralization rates below 10 °C appear to differ only minimally (Thamdrup et al., 2007; Scholze et al., 2020)."

**(21)      L389 – No mention about the biomass of benthic invertebrates which may adapt to respond to the variable MAR rates at the two fjords – i.e. the increase in MAR at Nuup Kangerlua may fuel a higher biomass benthos which results in similar OCBRs within both fjords. The benthic biology may be important here?**

There is work in progress by colleagues at the Greenlandic Institute of Natural Resources on epibenthos biomass. However, no results are available yet. We therefore chose to leave it out in the paragraph regarding OC uptake by the food web to keep this section brief and only based on what is known. We did, however, add benthic biomass in the future research section (see below): line 585: Accurate carbon budget construction requires integrated knowledge of primary production, zooplankton grazing, pelagic and benthic biomass as well as pelagic and benthic mineralization rates (Spilling et al., 2019), which are currently limited or lacking for these fjord systems.

**(22)        A section on Recommendations for further research would be helpful.**

We applied your comment and added the following section (lines 576-592):

**4.4 Recommendations for future research**

The expected link between elevated surface primary production in MTG-influenced fjords and OCBR was not observed. Future studies should therefore examine the mechanisms controlling this mismatch between pelagic productivity and sediment burial. In addition, our results imply that glacial influence is not necessarily the most important factor steering OCBR, which means that more Greenland fjord systems should be studied to better understand the effect of retreating MTGs on OC burial. Based on our results we identified the following avenues for future research:

1. Mass accumulation rates and OCBRs need to be studied in Greenlandic and other Arctic fjords, ideally applying the CRS method, for standardized comparisons. As not all of our MARs could be determined by the CRS method, these estimates should be verified in the future.

2. Accurate carbon budget construction requires integrated knowledge of primary production, zooplankton grazing, pelagic and benthic biomass as well as pelagic and benthic mineralization rates (Spilling et al., 2019), which are currently limited or lacking for these fjord systems. These parameters help quantify the mismatch between OC production and burial, which may arise from lateral transport processes or from OC incorporation into higher trophic levels. To address this, a more comprehensive understanding of food web dynamics and carbon flow in both fjords is essential.

3. An understanding of benthic OC cycling is important for quantifying carbon turnover at the sediment-water interface, potentially revealing processes that drive differences in OC burial efficiency in different fjord systems.

**(23)        The discussion and conclusion rely heavily on hypothetical scenarios due to a lack of data/observations for what is a very complex system. I appreciate the thought that has gone into trying to explain the results found within this study, however more data would be needed to do this. I think more could be done to look at the spatial distribution of sediments within the fjord systems as this can help to better understand hydrodynamics of the system as well as the benthic communities and carbon processing. There are more questions than answers resulting from this study, which is fine, however the authors may wish to refine their discussion to focus on what could be understood/ concluded from the results and provide recommendations for Further Work/ Research to build a stronger picture.**

Applying your feedback, we reformulated large parts of the discussion and reorganized the discussion to the following structure:

- **4.1 Surface sediment OC content** with a sub-section on **OC origin** (4.2.1), putting more emphasis on what can be derived from our data.
- **4.2 Organic carbon burial rates**, which was set as the 2nd sub section instead of the last paragraph. We also included primary production values from literature.
- We assembled our 3 main hypotheses in sub-sections within **4.3 Pelagic and geomorphological influence on OC burial:**
  - **4.3.1 OC preservation conditions**, which was condensed and more focused on the bottom water temperature differences we measured between the fjords. We removed the hypothesis regarding a difference in OC preservation related to Fe and Mn in both fjords from the discussion due to limited data to support this hypothesis.
  - We condensed section **4.3.2 Transport dynamics.** This section was included to provide a more nuanced view of MTG-induced upwelling, acknowledging that nutrient enrichment may also originate from upwelling at the fjord mouth, potentially influencing the patterns observed in our study. Regarding your remark on sediment distribution, we rephrased lines 472-477: "The slightly coarser grain size at Ameralik's mid and outer stations, despite their distance from glacial input, may indeed reflect input from the entrance sill. Furthermore, the topography of Ameralik with the deep depression behind the sill can promote downslope transport and sediment accumulation, resulting in the relative higher TOC and pigment content at AM5 (Hargrave and Nielsen, 1976; Wassmann et al, 1984; Erlandsson, 2008). Therefore, hydrodynamics, downslope transport, or a combination of both can decouple surface productivity from local sediment deposition."
  - We condensed the **4.3.3 Food web OC uptake** section.
- We incorporated your suggestion to add a section on Recommendations for future research. (As mentioned in (22)).

To avoid an overly long rebuttal letter, we refer the reviewer to our rewritten discussion in the manuscript. (From line 371 onwards).

**Conclusion**

 **(24)  L399 – There isn't enough evidence to suggest that MTGs DO function as carbon pumps – Perhaps change wording to COULD function as carbon pumps.**

Thank you for your insight. We replaced "do" with "could".

**Technical Corrections:**

 **(25)  L125 – TOC and TN not defined yet.**

Thank you for noticing this error. We therefore changed line 143: "At each station, three deployments were carried out for granulometry, pigment, total organic carbon (TOC) and total nitrogen (TN) analysis …"

 **(26)  L127 – grammatical point – fewer stations rather than less stations. Although Table 1 shows that there were the same number of stations sampled in both years?**

We adjusted line 145: Fewer sediment stations were sampled in 2021 compared to 2022; however, bottom water temperature measurements were obtained in both years (Table 1).

**(27)      L140 – text missing – change to 'conforming to the Wentworth scale'**

Changed to Line 168: Grain size fractions were classified according to the Wentworth scale (1922) as clay (< 4 µm), silt (4 – 63 µm), and sand (63 – 500 µm).

**(28)      Figure 4 – Can the values for the terrestrial and marine end members be shown – the range for 15N for marine is very broad – is that right?**

Thank you for your suggestion. We have therefore revised Figure 4 (line 356) including other remarks of other reviewers. We improved this figure by:

- Enlarging all elements of the figure.
- Adding a second panel, displaying δ13C against C:N ratio. We indicated the marine and terrestrial end-member δ13C values used in this study to determine the proportion of marine and terrestrial OC.
- Colors referring to Nuup Kangerlua and Ameralik were adapted to follow the color code of the map (Figure 1; line 122).
- In panel (a), we extended the marine and terrestrial ranges of panel (a) from δ13C -23 ‰ to δ13C -24‰ and from δ13C -25 ‰ to δ13C -21‰ (following Lamb et al., 2006), respectively.

[Figure]

**Figure 4. (a)** δ13C (‰ deviations from V-PDB) values plotted against δ15N (‰ deviations from air) values of the POM present in the sediment for the different station of Ameralik (filled symbols) and Nuup Kangerlua (open symbols). Typical marine and terrestrial ranges of δ13C (lamb et al., 2006) and δ15N (Zaborska et al., 2018) are indicated with rectangles. **(b)** δ13C plotted against C:N ratios. Ranges of marine and freshwater POC, and C3 terrestrial plants are displayed as rectangles for reference (values taken from Lamb et al., 2006). Marine (M) and terrestrial (T) δ13C end-members used in this study are indicated with arrows.

**(29)      Figure 5 – Data in blue and yellow represent the MTGs and LTGs respectively (red used in error). The mixed type is Red. Where have all the Greenland data points come from in 5b between 70-75N?**

We adjusted the figure caption by specifying the sources from which we extracted the data: line 404:

[Figure]

**Figure 5: (a)** TOC content of surface sediments along latitude. Data compiled from Smith et al. (2002), Thamdrup et al. (2007), Koziorowska et al. (2015), Cui et al. (2016a); Faust and Knies (2019), Włodarska-Kowalczuk et al. (2019), Laufer-Meiser et al. (2021) and this study. **(b)** Fraction of TOC of marine origin along latitude. Data compiled from Koziorowska et al. (2015), Faust and Knies (2019) and this study. Both figures display data from fjords located in high latitude countries: Alaska, Greenland, Norway and Svalbard. The grey band constraints the Greenland fjords investigated in this study. Data indicated in red and yellow represent Marine terminating-glacier (MTG) and land terminating-glacier (LTG)-influenced fjord systems, respectively. The mixed type represents fjords where the dominance of MTG(s) vs LTG(s) on the fjord's hydrology could not be differentiated from literature or satellite images are depicted in blue. Non-glacial fjords are represented in black. Both graphs were created following and updating the example of Faust and Knies (2019).

**Figure 6 – write out the word transport**

We applied your comment in figure 6 and changed some colours. (Line 557).

[Figure]

**Figure 6:** Schematic cross-sectional view of current regime and possible ways of phytoplankton or OC flow during summer in Nuup Kangerlua **(a)** and Ameralik **(b)** fjord systems. Tidal mixing above the sill area, estuarine circulation and intermediate baroclinic circulation occurs in both fjord systems, while the presence of MTGs in Nuup Kangerlua drives subglacial circulation through subglacial discharge. Nutrients are brought to the euphotic zone via tidal mixing and subglacial circulation. Turbid plumes, indicative of suspended sediment and organic matter input from glacier discharge and river runoff, are represented by the shaded brown texture. Green arrows represent phytoplankton or OC transport and remineralization of organic carbon at the sediment-water interface. A wider arrow points to higher expected flows. Station locations are marked along the fjords. The current dynamics illustrated for Nuup Kangerlua are based on Mortensen et al. (2018) and Stuart-Lee et al. (2023), while those for Ameralik are derived from Stuart-Lee et al. (2021; 2023).

**(30)      A2 – dots are grey, not orange.**

Thank you for noticing this error. We adjusted the figure caption to: "… Grey and black colors represent end of summer 2021 and spring 2022, respectively. Data from the two seasons is available for stations NK12, NK7, AM10 and AM5." (Line 661).

---

## Author Comment (AC3)

We are grateful to you for the time you invested in providing detailed and helpful feedback. Your comments have been invaluable in refining and strengthening our manuscript.

As a general clarification, we updated the station names for Nuup Kangerlua from GF6, GF7, etc., which referenced the Danish name Godthåbsfjord, to NK6, NK7, etc., to more clearly reflect their association with Nuup Kangerlua and enhance consistency throughout the manuscript.

We therefore added:

Line 137: It is important to note that earlier studies (e.g., Mortensen et al., 2011, 2014, 2018; Meire et al., 2015, 2017; Stuart-Lee et al., 2021, 2023) referred to the same stations in Nuup Kangerlua using the prefix "GF", derived from the Danish name *Godthåbsfjord*. In this study, we use the prefix "NK" instead, to reflect the Greenlandic name *Nuup Kangerlua*.

To avoid confusion, we updated this naming (NK) in your comments as well. We also numbered your questions and highlighted them in bold to clearly structure our responses.

**Introduction**

    **(1) Line 27 states that fjords OC burial contributes to around 10% of blue carbon burial. The work of Smith et al. (2015) states that fjord OC accumulation is equivalent to 10% of carbon buried in other marine sedimentary systems which does not include blue carbon environments (saltmarsh, seagrass and mangroves).**

Thank you for the clarification. You are correct, Smith et al. (2015) compares fjord carbon burial to other marine sedimentary systems, not blue carbon environments. We have accordingly replaced "blue carbon" with "carbon" to reflect this more accurately.

Line 35: Fjord systems play a crucial role in burial and long-term storage of organic carbon (OC), contributing to approximately a tenth of the annual carbon burial (Smith et al., 2015).

In addition, we adjusted the Abstract: line 14: Fjord systems play a crucial role in the burial and long-term storage of organic carbon (OC).

    **(2) Line 45 – refrain from using "references therein". Cite the relevant studies.**

We added references: line 56: Since MTGs terminate in the ocean, this sub-glacial meltwater rises up from the bottom of the glacier within the fjord basin entraining nutrients present in deeper water layers (Meire et al., 2017; Hopwood et al., 2018; 2020; Kanna et al., 2018; Cape et al., 2019; Halbach et al., 2019; Seifert et al., 2019).

    **(3) Lines 40 – 51 – In this paragraph you discuss how the different types of glaciers enhance or reduce nutrients entering the fjords. Which nutrients?**

Following your remark, we adjusted the text and specified the nutrients: line 56 - 60: Since MTGs terminate in the ocean, this sub-glacial meltwater rises up from the bottom of the glacier within the fjord basin entraining nutrients, like nitrate, ammonium and phosphate, present in deeper water layers (Meire et al., 2017; Hopwood et al., 2018; 2020; Kanna et al., 2018; Cape et al., 2019; Halbach et al., 2019; Seifert et al., 2019). This upwelling water mass replenishes essential nutrients for primary production in the surface waters, crucial for sustaining phytoplankton proliferation beyond the initial spring bloom phase.

    **(4) Line 66 – Throughout the manuscript the term "carbon sequestration" is used. Sequestration is when carbon is removed from the atmospheric pool. Fjords are donor**

**environments (Middleburg, 2019) which means the receive and store carbon sequestered in other environments (terrestrial or marine). Accumulates or buries are more accurate terms to use when describing these processes.**

Thank you for pointing this out. We agree that "carbon sequestration" is not the most accurate term in this context. Following your suggestion, we have replaced 'sequestration' with 'burial' throughout the text.

**(5) Lines 81-85 – This line does not add anything to the introduction, please remove.**

We removed the last part of the introduction, which was originally: "Therefore, we compared carbon storage and burial in two neighbouring, sub-Arctic fjord systems which both feature a sill at their entrance and are subjected to similar offshore currents and similar geology in their catchments, but have a different glacier influence (MTG-dominated vs. LTG-dominated fjords)." The introduction runs until line 95.

**Methodology**

**(6) Line 104 – is Stuart Lee et al., 2021 correctly referenced.**

Regarding line 116: "Being more than twice as shallow compared to the entrance sills in Nuup Kangerlua, the sill restricts inflow of relatively warmer and more saline sub-polar mode water (SPMW), resulting in relatively lower bottom water temperatures of ~0–1°C (spring and summer 2019 data; Stuart Lee et al., 2021)."

Stuart-Lee et al. (2021) states: "A comparison of deep water properties reveals another important difference between these two fjords. Between 50 and 400 m, Ameralik was fresher (by 0.2) and cooler (by **between 0.7 and 1.1°C**) than Godthåbsfjord for each of the studied months (May, July and September). This difference also applied to the water **below 400 m**, as indicated by the gray rectangles in Figure 6. The **difference in deep water properties is related to the sill depth, which determines the inflow of coastal water and differs between the two fjords**." In which Figure 6 shows bottom water temperatures of Ameralik around 0°C and Nuup Kangerlua slightly above 1°C. Given these observations made by Stuart-Lee et al. (2021), we believe that this work fits as a reference. We agree that we also should have referred to our own water temperature measurements during the sampling period. We therefore adjusted the sentence:

Line 116: "Being more than twice as shallow compared to the entrance sills in Nuup Kangerlua, the sill restricts inflow of relatively warmer and more saline sub-polar mode water (SPMW), resulting in bottom water temperatures below 1°C (Stuart- Lee et al., 2021; Table 1).

Table 1 can be found at line 152:

**Table 1.** Sampling dates, coordinates, water depth, and bottom water temperatures (BWT) of sampled stations in Nuup Kangerlua (NK) and Ameralik (AM).

| Station | Date(s) sampled | Longitude (N) | Latitude (W) | Depth (m) | BWT (°C) 2021 | 2022 |
|---|---|---|---|---|---|---|
| NK13 | 31/05/2022 | 64° 40.8 | 50° 17.3 | 476 | 1.47 | 1.41 |
| NK12 | 31/08/2021 20/05/2022 | 64° 42.9 | 50° 32.8 | 531 | 1.41 | 1.35 |
| NK10 | 31/08/2021 | 64° 36.6 | 50° 57.5 | 579 | 1.32 | 0.81 |
| NK9 | 24/05/2022 | 64° 33.0 | 51° 0.9 | 602 | 1.23 | 0.67 |
| NK7 | 01/09/2021 20/05/2022 | 64° 25.5 | 51° 3.4 | 626 | 1.29 | 0.64 |
| NK6 | 30/08/2021 | 64° 22.0 | 51° 0.4 | 630 | 1.28 | 0.62 |
| AM10 | 02/09/2021 18/05/2022 | 64° 11.0 | 50° 25.9 | 350 | 0.49 | 0.45 |
| AM8 | 18/05/2022 | 64° 10.4 | 50° 45.3 | 488 | 0.59 | 0.56 |
| AM5 | 03/09/2021 24/05/2022 | 64° 05.7 | 51° 11.3 | 730 | 0.56 | 0.59 |

**(7) 124 – Solid phase – would sediment be a better description. I don't understand the need for this as there is no liquid phase sampling.**

We changed the paragraph into: "2.3 Sediment analysis" (line 157). There was originally a liquid phase sampling section, but it was omitted from this paper.

**(8) Line 125 – the n=3 and n=1 is not required as you state in the text there were 3 deployments for geochemistry and sedimentology and 1 for dating.**

We removed n=3 & n=1.

**(9) Line 128 – remove "thick"**

We removed "thick".

**(10)        Line 129 – expand on why the sampling intervals was increased from 1 to 2 cm downcore. Would it not have been more useful to be consistent throughout.**

Slices of 2 cm thickness were collected at beyond 10 cm depth to account for the decrease in [210]Pb activity with sediment depth, ensuring sufficient material for reliable detection above background levels.

We adjusted the sentence, line 148: "Sediment intended to derive sediment accumulation rates ([210]Pb analysis) was further sliced beyond 10 cm in intervals of 2 cm until the end of the core (ranging from 10 to 44 cm sediment) ensuring sufficient material for reliable [210]Pb activity detection above background levels."

**(11)        Table 1 – reduce the number of decimals points the BWT is reported to. " decimal points will be suitable.**

We adjusted Table 1 (line 152):

**Table 1** Sampling dates, coordinates, water depth, and bottom water temperatures (BWT) of sampled stations in Nuup Kangerlua (NK) and Ameralik (AM).

| Station | Date(s) sampled | Longitude (N) | Latitude (W) | Depth (m) | BWT (°C) 2021 | BWT (°C) 2022 |
|---------|------------------|----------------|---------------|------------|---------------|---------------|
| NK13 | 31/05/2022 | 64° 40.8 | 50° 17.3 | 476 | 1.47 | 1.41 |
| NK12 | 31/08/2021 20/05/2022 | 64° 42.9 | 50° 32.8 | 531 | 1.41 | 1.35 |
| NK10 | 31/08/2021 | 64° 36.6 | 50° 57.5 | 579 | 1.32 | 0.81 |
| NK9 | 24/05/2022 | 64° 33.0 | 51° 0.9 | 602 | 1.23 | 0.67 |
| NK7 | 01/09/2021 20/05/2022 | 64° 25.5 | 51° 3.4 | 626 | 1.29 | 0.64 |
| NK6 | 30/08/2021 | 64° 22.0 | 51° 0.4 | 630 | 1.28 | 0.62 |
| AM10 | 02/09/2021 18/05/2022 | 64° 11.0 | 50° 25.9 | 350 | 0.49 | 0.45 |
| AM8 | 18/05/2022 | 64° 10.4 | 50° 45.3 | 488 | 0.59 | 0.56 |
| AM5 | 03/09/2021 24/05/2022 | 64° 05.7 | 51° 11.3 | 730 | 0.56 | 0.59 |

**(12)      Line 139 – 153 needs significantly expanded; not enough information is provided to replicate the study and must be supported by references.**

We agree that this section was too concise and did not provide sufficient detail for reproducibility. Thank you for highlighting this and for your valuable suggestions in comments 13–17, which have helped guide our revisions. We have substantially expanded the Materials and Methods section and included additional references where appropriate.

**(13)      Line 139 – How was the sediment prepared for analysis. Was there any preparation such as removal of organics or carbonate using Hydrogen peroxide and HCl. Was any detergent (i.e. Calgon) used to prevent flocculation of the clays?**

We adjusted this analysis description to improve clarity: lines 159 – 169: Grain size distribution was determined on oven-dried sediment samples (dried at 60 °C for at least 48 hours). After homogenization, coarse material > 2 mm was removed by sieving. A subsample of 0.1 – 1 g was resuspended in water and analyzed using a Malvern Mastersizer 2000 with the Hydro 2000S module (size range: 0.02 – 2000 μm), which operates based on laser diffraction. The sample is sonicated for 60 s to prevent flocculation of clays before it goes through the laser. No pretreatment to remove organic or inorganic carbon was performed prior to analysis. Grain size fractions were classified according to the Wentworth scale (1922) as clay (< 4 μm), silt (4 – 63 μm), and sand (63 – 500 μm).

**(14)      Line 140 states Porosity was obtained by dividing the porewater volume by the wet sediment volume. How was the porewater volume calculated?**

We elaborated on the porewater analysis: lines 171 – 176: Sediment porosity was estimated gravimetrically using a modified water displacement method. A pre-weighed 10 mL graduated measuring cylinder was filled with 2 – 5 g of homogenized wet sediment. After careful addition of ultrapure water (Milli-Q) to the 10 ml mark (measured by the lower meniscus), the cylinder was weighed again. The sample was then dried at 80 °C for ~48 h and reweighed. Porosity ($\phi$) was calculated based on the difference between the wet sediment weight and the dry sediment weight (i.e., the mass of porewater), recalculated to pore water volume through correction for salinity and divided by the estimated volume occupied by the sediment (calculated as 10 ml minus the volume of water added).

**(15)    Line 141 – What temperature were the samples dried and for how long? + Line 143 states "After decalcification with 37% HCl, total organic carbon (TOC) was also measured" – Was this acid fumigation or direct acidification. I assume it is acid fumigation as 37% conc, HCl would not be used for direct acidification.**

Your comments were addressed in the following rewritten lines (lines 179 - 188):

To determine total carbon (TC), total organic carbon (TOC), and total nitrogen (TN), sediment samples were oven-dried at 60 °C for 48 hours ground using mortar and pestle, and homogenized. Between 20 – 35 mg of dried and homogenized sediment was weighed and placed into pre-weighed silver cups. For TC and TN analysis, silver cups were sealed by folding with tweezers into compact spheres to ensure complete combustion. For TOC analysis, carbonate removal was performed by stepwise acidification: 2 – 3 drops of increasing concentrations of HCl (1%, 2%, 5%, and 10%) were added sequentially using a glass pipette. After each addition, samples were dried at 60 °C for 1 to 2 hours. This procedure was repeated over 2 – 3 days until no bubbling was observed and carbonate removal was confirmed. All measurements were conducted using a Flash 2000 NC Sediment Analyzer (Interscience), which quantifies carbon and nitrogen via dynamic flash combustion and chromatographic separation. From these data, the molar C:N ratios were calculated dividing TOC by TN and inorganic carbon (IC) was determined by subtracting TOC from TC.

**(16)    Line 145 – What does "see further" refer to? + Line 146 – for the stable isotope analysis provide a preparation method, I assume the same as the EA.**

We specified the section to which we were referring and provided more details on the stable isotope analysis in lines 188 – 191: To investigate the origin of the organic matter (see 2.3.1), stable isotope composition, $\delta^{13}C$ (‰ deviations from V-PDB) and $\delta^{15}N$ (‰ deviations from air), was measured with an elemental analyzer (Thermo Flash EA1112 elementanalyzer) coupled to an isotope ratio mass spectrometer (Thermo Finnigan Delta V, IRMS). Prior to analysis, the same steps were followed as for TOC and TN analysis, except samples were freeze dried.

**(17)    Line 151 – Briefly expand on the methodology of Wright and Jeffrey (1997), how were the pigments extracted and what tool was used to measure them?**

We adjusted the text and provided more information on the pigment analysis in lines 197 - 210: To explore how glacier type affects marine primary productivity and whether and how it is incorporated in the sediment, we additionally measured, for each sediment slice, the content of chlorophyll-a (Chl-a) and of its degradation products (pheophorbide-a, and pheophytin-a, pheophorbide-a like, and pheophytin-a like following Wright and Jeffrey (1997). For pigments extraction, 2 ml acetone (90%) was added to 0.5 g freeze dried sediment under red light conditions preventing pigment degradation. The samples were subsequently sonicated for 30 s and incubated overnight at 4 °C in the dark to aid pigment release.

Afterwards, the samples were centrifuged (10 min, 4000 rpm, 4 °C) and the supernatant was passed through 0.2 µm PTFE filters. Pigment separation was performed using an HPLC system (Agilent 1200 Infinity II, Agilent Technologies) equipped with a cooled auto-sampler, column oven, photodiode array detector, and fluorescence detector, following the method of Van Heukelem and Thomas (2001). Chlorophyll a and its degradation products were identified at 665 nm wavelength. Individual pigment concentrations were determined using the response factors of the respective standards. The ratio of Chl-a to Chloroplastic Pigment Equivalent (CPE, comprising the sum of all aforementioned pigments) was used as a proxy for the "freshness" or lability of photosynthetically produced organic matter (Schubert et al., 2005; Koho et al., 2008).

> **(18)** **Line 185 – Both the linear and CRS calculation methods were used to determine the OCAR for a mix of sites. Please state why two different methods were used. I understand the Cs peaks allowed the CRS to be used on some cores, but would it not been useful for comparison to use the linear method on all cores.**

In a very dynamic environment such as fjords with mass discharge from glaciers, we suppose that the CRS method offers a higher chance to describe the sedimentation processes. This model gave good geochronology in the cores shown in figure A3, where every $^{210}$Pb age is confirmed by $^{137}$Cs measurements: the average sedimentation rate calculations in AM5, AM8, NK7 and NK9 are the most robust estimates, as an additional independent tracer ($^{137}$Cs) could be used.

Vice versa, the absence of a clear $^{137}$Cs peak in the other stations did not allow to confirm the CRS dating and, since we needed an estimate of sedimentation rate, we applied the CF:CS model. Based on the available data, we could not get a more accurate dating for stations NK10, NK12, NK13 and AM10. It would be indeed better to emphasize that MAR and SAR values of stations NK10, NK12, NK13 and AM10 based on CF:CS dating are estimates that should be verified in future research. Therefore, we adapted the "2.4.1 Organic carbon burial rate" section in the Materials and Methods section:

Lines 261-272: "Sedimentation rates at stations AM5, AM8, NK7, and NK9 were determined using the constant rate of supply (CRS) model (Appleby, 2001), as a distinct increase in $^{137}$Cs was detected at these sites (Fig. A1), supporting the CRS-based chronology. The observed increase in $^{137}$Cs activity is attributed to global fallout from atmospheric nuclear weapons testing, which peaked in 1963. In contrast, the CF:CS (constant flux:constant sedimentation) model (Sanchez-Cabeza and Ruiz-Fernández, 2012) was applied to stations NK10, NK12, NK13, and AM10, where the $^{210}$Pb$_{ex}$ profiles exhibited approximately exponential trends but lacked a clearly defined $^{137}$Cs peak. For these stations, log-transformed $^{210}$Pb$_{ex}$ activities were plotted against cumulative dry mass depth (g cm$^{-2}$) for each station. As a result, the sedimentation rate estimates for these stations should be treated with caution and verified in future studies. Mass accumulation rates (MAR, kg m$^{-2}$ yr$^{-1}$) were derived from the slope of the linear regression (for CF:CS) or from the CRS model output. Bulk sediment accumulation rates (SAR, mm yr$^{-1}$) were calculated by dividing MAR by the average bulk density at each station. Organic carbon burial rates (OCBR) were then calculated by multiplying MAR by the TOC content at the 9 – 10 cm sediment layer. …"

> **(19)** **Line 187 – State which year you are assigning to the 137Cs peak, Chernobyl or weapons testing?**

We agree that this should have been stated in the text. The following sentence was added to the Material and Methods section to clarify that the $^{137}$Cs peaks were linked to the nuclear weapons testing:

: The observed increase in [137]Cs activity is attributed to global fallout from atmospheric nuclear weapons testing, which peaked in 1963.

**(20)      Line 187 states there are clear 137Cs peaks in cores NK9, NK7, AM8 and AM5. From figure A1 that these peaks could be heavily distorted by mixing how confident are you these are unimpacted records.**

See joint answer after next comment.

**(21)      Line 193 states "No bioturbated or mixed upper layer was observed in the profiles". The radionuclide profiles would suggest significant mixing, which the authors later discuss in the text. There suitability of these cores for dating and the quality of the resultant outputs must be discussed.**

As you correctly point out, we should have been more clear on why we think bioturbation is limited. The same concern was expressed by another reviewer. Therefore, we added lines 281-288: "We did not apply corrections for bioturbation or mixing processes, as the $^{210}Pb_{ex}$ profiles do not show evidence of such activity in the upper sediment layers. However, these processes cannot be conclusively ruled out, particularly since some of the $^{137}$Cs profiles feature broad activity peaks. Nonetheless, the $^{210}$Pb-derived chronology appears to be supported by the $^{137}$Cs profiles in AM5, AM8, NK7 and NK9 (Smith, 2001; Barsanti et al., 2020). The broad $^{137}$Cs curves or inflections, marking sustained elevation in $^{137}$Cs activity after an initial increase followed by a gradual decrease over time, are therefore more likely explained by continued exposure of settling particles to residual $^{137}$Cs in the overlying water after 1963. As a result, younger sediment layers also contain measurable amounts of $^{137}$Cs, smearing the signal across multiple horizons. This phenomenon has been observed in other marine settings (Tamburrino et al. 2019) and even in lake sediments (Drexler et al., 2018).

Following references were added in the reference list:

Barsanti, M., Garcia-Tenorio, R., Schirone, A., Rozmaric, M., Ruiz-Fernández, A. C., Sanchez-Cabeza, J. A., and Osvath, I.: Challenges and limitations of the $^{210}$Pb sediment dating method: Results from an IAEA modelling interlaboratory comparison exercise, Quaternary Geochronology, 59, 101093, https://doi.org/10.1016/j.quageo.2020.101093, 2020.

Drexler, J. Z., Fuller, C. C., and Archfield, S.: The approaching obsolescence of $^{137}$Cs dating of wetland soils in North America, Quaternary Science Reviews, 199, 83–96, https://doi.org/10.1016/j.quascirev.2018.08.028, 2018.

Smith, J. N.: Why should we believe $^{210}$Pb sediment geochronologies?, Journal of Environmental Radioactivity, 55, 121–123, https://doi.org/10.1016/S0265-931X(01)00110-2, 2001.

Tamburrino, S., Passaro, S., Barsanti, M., Schirone, A., Delbono, I., Conte, F., Delfanti, R., Bonsignore, M., Del Core, M., Gherardi, S., en Sprovieri, M.: Pathways of inorganic and organic contaminants from land to deep sea: The case study of the Gulf of Cagliari (W Tyrrhenian Sea), The Science Of The Total Environment, 647, 334–341, https://doi.org/10.1016/j.scitotenv.2018.07.467, 2018.

**Regarding the suitability of the cores for dating**, we chose to omit the sedimentation rate estimate of NK6, because (1) it is not a deep profile; only 10 cm deep and (2) $^{210}Pb_{xs}$ activity is indeed very low. This may be due to erosion conditions at the sill slope where older sediment material (characterized by low $^{210}Pb_{ex}$ activity) surfaces due to removal of younger sediment or older sediment has been transported to this location, which is also visible in NK7 and 9 profiles. The omission of the low MAR estimate at

station NK6 resulted in a slightly higher average organic carbon burial rate for Nuup Kangerlua: $18.0 \pm 1.6$ g m$^{-2}$ yr$^{-1}$ versus the previously calculated $14.1 \pm 1.6$ g m$^{-2}$ yr$^{-1}$.

**Results**

**(22)**      **Line 220 – What is the reason for such shifts in porosity and dry bulk density.**

Taking into account your comment, we adjusted the description in the results section: lines 323-327: Porosity and dry density generally fluctuated with sediment depth without a consistent pattern across most stations. In contrast, station NK10 exhibited the expected trend of decreasing porosity and increasing dry density with depth. These variations appeared to be influenced by grain size, although the processes driving the trends at NK10 are less clearly linked to sediment texture.

The related figure can be found further below at question (25).

**(23)**      **Line 37 – remove "while"**

At line 37, there was no "while" within the sentence, but we removed the redundant "while" in line 344 by splitting the sentence in two: While the $\delta^{13}$C value fluctuated widely at NK13 ranging from -26.3 to -23.8 ‰, indicating a stronger terrestrial influence and a more heterogeneous mixture of organic matter sources. Notably, $\delta^{15}$N at this station increased consistently with depth, from 5.7 ‰ to 12.2 ‰.

**Figures**

**(24)**      **Figure 1 – Add site numbers to the map. Change colour of marker for Nuuk it is currently too close to the sampling site symbol. Define the inner, middle and outer sections of the fjord on the maps.**

We improved Figure 1 (line 122) based on your comment and the other two reviewers by:

- Adding a legend, depicting stations sampled in Nuup Kangerlua and stations in Ameralik in a different color.
- Adding station labels and adjusting the symbol of Nuuk to differentiate it from the station symbols.
- Situating the outer, mid and inner zone in the (b) panel.
- Enlarging (a) and (b) panel.

[Figure]

**Figure 1. (a)** Map showing sampling locations in Nuup Kangerlua (fed by three marine-terminating glaciers and three land-terminating glaciers) and Ameralik (receiving meltwater from a land-terminating glacier). Greenland Ice Sheet (GrIS) is depicted in white. **(b)** Water depth profiles along-axis (white dashed lines) Nuup Kangerlua (top) and Ameralik (bottom). Both fjord basins are divided in an outer, mid and inner section behind the entrance sill(s).

**(25)** **Figure 2 has dry bulk density data. But this is not mentioned in the method section. Lines joining the points would be useful as it is hard to follow some of the downcore data (i.e., NK10).**

We added dry density to the M&M section: Line 171: Sediment porosity was estimated gravimetrically using a modified water displacement method. A pre-weighed 10 ml graduated measuring cylinder was filled with 2 – 5 g of homogenized wet sediment. After careful addition of ultrapure water (Milli-Q) to the 10 ml mark (measured by the lower meniscus), the cylinder was weighed again. The sample was then

dried at 80 °C for ~48 h and reweighed. Porosity (ϕ) was calculated based on the difference between the wet sediment weight and the dry sediment weight (i.e., the mass of porewater), divided by the estimated volume occupied by the sediment (calculated as 10 ml minus the volume of water added). Dry bulk density was obtained by dividing dry mass by bulk volume.

We added horizontal grid lines to Figures 2 and 3 to enhance visual clarity and support more accurate comparison of values along the y-axis.

[Figure]

**Figure 2.** Sediment profiles of median grain size (µm), porosity and dry density (g cm$^{-3}$) of Nuup Kangerlua (GFNK stations) and Ameralik (AM stations). Error bars represent SD (n = 3 for NK6, NK9, NK13 and AM8 and n = 6 for NK7, NK10, NK12, AM5 and AM10). Only one replicate for porosity and dry density.

**(26)     Could Figure 2 and 3 be combined.**

We understand that all data placed next to each other improves comparison of parameters, but we decided to keep both figures separate as too much different parameters within the same figure decreases clarity.

**(27)     Figure 4 – add the end-member values used in the study to the plot. Also, it would be useful to plot d$^{13}$C vs CN**

Thank you for this insightful suggestion. We agree that plotting δ$^{13}$C against C:N ratios adds value, as both are commonly used to distinguish between terrestrial and marine organic carbon sources. We have therefore revised Figure 4 (line 342) including other remarks of other reviewers. We improved this figure by:

-     Enlarging all elements of the figure.

- Adding a second panel, displaying $\delta^{13}$C against C:N ratio. We indicated the marine and terrestrial end-member $\delta^{13}$C values used in this study to determine the proportion of marine and terrestrial OC.
- Colors referring to Nuup Kangerlua and Ameralik were adapted to follow the color code of the map (Figure 1; line 122).
- In panel (a), we extended the marine and terrestrial ranges from $\delta^{13}$C -23 ‰ to $\delta^{13}$C -24‰ and from $\delta^{13}$C -25 ‰ to $\delta^{13}$C -21‰ (following Lamb et al., 2006), respectively.

[Figure]

**Figure 4. (a)** $\delta^{13}$C (‰ deviations from V-PDB) values plotted against $\delta^{15}$N (‰ deviations from air) values of the POM present in the sediment for the different station of Ameralik (filled symbols) and Nuup Kangerlua (open symbols). Typical marine and terrestrial ranges of $\delta^{13}$C (lamb et al., 2006) and $\delta^{15}$N (Zaborska et al., 2018) are indicated with rectangles. (b) $\delta^{13}$C plotted against C:N ratios. Ranges of marine and freshwater POC, and C3 terrestrial plants are displayed as rectangles for reference (values taken from Lamb et al., 2006). Marine (M) and terrestrial (T) $\delta^{13}$C end-members used in this study are indicated with arrows.

In addition we added to the text, section "2.3.1 Calculation of marine organic carbon fraction": lines 212 – 219: Stable isotope composition in addition to C:N ratios of settled organic matter in fjord sediments has been used in multiple studies to estimate the proportion of marine versus terrestrially derived organic matter (St-Onge and Hillaire-Marcel, 2001; Koziorowska et al., 2015; Smeaton & Austin, 2017; Zaborska et al., 2018; Faust and Knies, 2019; Limoges et al., 2020; Placitu et al., 2024). Terrestrial organic matter, primarily derived from vascular plants, tends to have higher C:N ratios (> 12) and more depleted $\delta^{13}$C values (-25 to -30 ‰ $\delta^{13}$C) due to the dominance of lignin-rich, cellulose-based material and the use of C3 photosynthesis pathways (Lamb et al., 2006). In contrast, marine organic matter, originating from phytoplankton and other aquatic organisms, typically shows lower C:N ratios and less negative $\delta^{13}$C values (-18 to -24 ‰  $\delta^{13}$C), reflecting a protein-rich composition and different carbon fixation mechanisms (Lamb et al., 2006).

**(28)       Table 2 – in the caption state which model was used to determine these rates.**

Following your remark, we revised the caption of the table: Line 369: **Table 2.** Mass accumulation rate (MAR), sediment accumulation rate (SAR), and organic carbon burial rate (OCBR) per station. The CRS method was applied at stations NK7, NK9, AM5, and AM8, while the CF:CS method was used for stations NK10, NK12, NK13, and AM10. "NK" denotes Nuup Kangerlua and "AM" Ameralik.

**(29)** **Figure 2A – Colours missing, there should be orange and black according to the caption.**

Thank you for noticing our mistake. We changed the figure caption to: "… Grey and black colors represent end of summer 2021 and spring 2022, respectively. Data from the two seasons is available for stations NK12, NK7, AM10 and AM5." (Line 662).

**Discussion**

**(30)** **I also believe that the discussion needs to be reworked. This section is largely focused on reasons why the results do not match what was predicted. The study goes down multi avenues of enquiry but these are largely subjective as there is no data available. Much of this could be condensed into a single section opposed to being four sub-sections.**

Applying your feedback, we reformulated large parts of the discussion and reorganized the discussion to the following structure:

- **4.1 Surface sediment OC content** with a sub-section on **OC origin** (4.2.1), putting more emphasis on what can be derived from our data.
- **4.2 Organic carbon burial rates**, which was set as the 2$^{nd}$ sub section instead of the last paragraph. We also included primary production values from literature.
- We assembled our 3 main hypotheses in sub-sections within **4.3 Pelagic and geomorphological influence on OC burial:**
  - o **4.3.1 OC preservation conditions**, which was condensed and more focused on the bottom water temperature differences we measured between the fjords. We removed the hypothesis regarding a difference in OC preservation related to Fe and Mn in both fjords from the discussion due to limited data to support this hypothesis.
  - o We condensed section **4.3.2 Transport dynamics.** This section was included to provide a more nuanced view of MTG-induced upwelling, acknowledging that nutrient enrichment may also originate from upwelling at the fjord mouth, potentially influencing the patterns observed in our study
  - o We condensed the **4.3.3 Food web OC uptake** section.
- We incorporated the suggestion of Anonymous Referee #1 to add a section on Recommendations for future research.

To avoid an overly long rebuttal letter, we refer the reviewer to our rewritten discussion in the manuscript. (From line 371 onwards).

---

## Author Comment (AC4)

**4 Discussion**

With this study we wanted to identify to what extent the higher surface water productivity in a fjord with a MTG is reflected in carbon burial potential of the deep water sediments. We therefore expected higher OC content and OCBRs in sediments of Nuup Kangerlua compared to Ameralik, as MTGs present in Nuup Kangerlua increase nutrient upwelling, allowing primary productivity to extend over longer periods. Indeed, earlier studies by Stuart-Lee et al. (2023) and Meire et al. (2023) noted comparable primary productivity in Nuup Kangerlua and Ameralik at the start of the productive season (April, May). Yet, with increasing meltwater discharge, a summer bloom was observed in Nuup Kangerlua which led to a greater overall phytoplankton biomass compared to Ameralik (Stuart-Lee et al., 2023; Meire et al., 2023). However, in this study, we found a higher OC content in sediments of outer and mid fjord stations AM5 and AM8 in Ameralik compared to Nuup Kangerlua. These findings are supported by observations from a gravity core sampled nearby station AM5, which also revealed similar elevated carbon content in the sediment (Møller et al., 2006). Our results therefore do not support the hypothesis of higher carbon burial potential of MTG fjords compared to LTG driven systems.

**4.1 Surface sediment OC content**

The OC content in the sediments of Nuup Kangerlua and Ameralik is representative for (sub-) Arctic fjord sediments (Fig. 5A). In terms of fresh organic matter, we found an average Chl-a content in Nuup Kangerlua's sediments which was slightly below the typical range observed in other North Atlantic fjords (Włodarska-Kowalczuk et al., 2019). In contrast, Ameralik exhibited an average Chl-a content nearly three times higher than the maximum values reported for Svalbard fjords (Włodarska-Kowalczuk et al., 2019). This elevated average is largely driven by the exceptionally high Chl-a content observed at station AM5.

So far, studies comparing MTG and LTG fjord systems are limited (Koziorowska et al., 2015; Laufer-Meiser et al., 2021). These studies suggest that MTG fjords tend to exhibit higher OC accumulation, as indicated by elevated OC content in surface sediments. However, when comparing the LTG system Ameralik and the MTG system Nuup Kangerlua with datasets from other fjords (Smith et al., 2002; Thamdrup et al., 2007; Koziorowska et al., 2015; Cui et al., 2016; Faust and Knies, 2019; Włodarska-Kowalczuk et al., 2019; Laufer-Meiser et al., 2021), we observed that surface sediment OC content in LTG and even non-glaciated fjords can be comparable to that of MTG systems across the (sub-)Arctic region (Fig. 5A). Nevertheless, it is important to note that LTG fjords are underrepresented in current datasets, and low-glacial-activity MTG systems may bias comparative interpretations.

These observations suggest that factors beyond glacial influence play a significant role in controlling the degree of benthic-pelagic coupling. Specifically, the presence of MTGs does not inherently result in higher OC accumulation within sediments compared to systems without subglacial upwelling. However, elevated MARs may dilute OC content with inorganic material, potentially skewing these observations. Additionally, higher TOC content in surface sediments does not automatically equate to more efficient OC burial.

[Figure]

**Figure 5: (a)** TOC content of surface sediments along latitude. Data compiled from Smith et al. (2002), Thamdrup et al. (2007), Koziorowska et al. (2015), Cui et al. (2016a); Faust and Knies (2019), Włodarska-Kowalczuk et al. (2019), Laufer-Meiser et al. (2021) and this study. **(b)** Fraction of TOC of marine origin along latitude. Data compiled from Koziorowska et al. (2015), Faust and Knies (2019) and this study. Both figures display data from fjords located in high latitude countries: Alaska, Greenland, Norway and Svalbard. The grey band constraints the Greenland fjords investigated in this study. Data indicated in red and yellow represent Marine terminating-glacier (MTG) and land terminating-glacier (LTG)-influenced fjord systems, respectively. The mixed type represents fjords where the dominance of MTG(s) vs LTG(s) on the fjord's hydrology could not be differentiated from literature or satellite images are depicted in blue. Non-glacial fjords are represented in black. Both graphs were created following and updating the example of Faust and Knies (2019).

**4.2.1 OC origin**

An important clue in resolving the observed patterns can be found in the deepest part of Ameralik's basin. There, specifically at station AM5, we measured a 5 times higher Chl-a content combined with 1.7 times higher Chl-a:CPE ratios compared to the maximum values in sediments of Nuup Kangerlua, which points to an enhanced preservation of fresh organic matter (i.e. more labile OC) within these sediments. The Chl-a content remained elevated throughout the entire 10 cm sediment profile and was consistent between spring and summer data. A difference in timing of the onset of the phytoplankton bloom between the two fjords, as previously observed (Stuart-Lee et al., 2023), could have led to an earlier build-up of pigments at the seafloor of Ameralik compared to Nuup Kangerlua at the time of sampling. However, the relatively elevated values throughout the 10 cm sediment profiles and the consistency between spring and summer data exclude such sampling time bias. In Svalbard, Koziorowska et al. (2015) also observed higher OC content in the surface sediments of a LTG-influenced fjord versus a MTG-impacted fjord. The LTG-fed fjord appeared to receive a higher fraction of terrestrial OC, which tends to be more resistant against degradation compared to marine OC (Wakeham and Canuel; 2006; Koziorowska et al., 2015). Yet, in our case, the sediment stable isotope composition and C:N ratios of both fjords reflect OC of predominantly marine origin in both fjords, likely due to the limited vegetation and a catchment geology consisting of orthogneisses, granodiorites and granites rather than organic-rich sedimentary rocks (Næraa et al., 2014) (Fig. 4; Fig. 5B). An

exception is inner station NK13 in Nuup Kangerlua, which displayed $\delta^{15}N$ of marine signature, though depleted $\delta^{13}C$ values which combined with C:N values indicated a freshwater provenance (Fig. 4b). Since elevated $\delta^{15}N$ values can also be caused by degradation (Dai et al., 2005), this station may contain OM more of terrestrial origin. In contrast, the stable isotope composition found at the head of Ameralik, in front of the land-terminating glacier, does not indicate a dominant terrestrial input.

So in general, the higher OC content in Ameralik sediments is not related to increased terrestrial input in the LTG fjord compared to the MTG-dominated fjord. In fact, sediments from both fjords receive OM from predominantly marine origin.

**4.2 Organic carbon burial rates**

Despite the higher OC content observed in the outer and mid part of the LTG-fed fjord, OCBRs were similar in both fjords due to the relatively higher MARs in Nuup Kangerlua. The higher MARs in Nuup Kangerlua result from the substantially higher discharge that three MTGs and three LTGs generate compared to the input of a single LTG in Ameralik. The average OCBR in Nuup Kangerlua was only on average slightly higher ($18.0 \pm 1.6$ g OC m$^{-2}$ yr$^{-1}$), but not significantly, compared to Ameralik ($16.2 \pm 1.7$ g OC m$^{-2}$ yr$^{-1}$). However, it must be noted that glacial run-off induced lithogenic dilution of OC can lead to an underestimation of OCBR in Nuup Kangerlua. Nevertheless, the observed values fall within the range of sub-Arctic fjords and Arctic fjords impacted by active glaciers (Włodarska-Kowalczuk et al., 2019).

On the local scale, Meire et al. (2023) estimated that annual pelagic primary production in 2016 was approximately three times higher in a head station of Nuup Kangerlua (~90 g C m$^{-2}$ yr$^{-1}$ at NK10) than in a head station of Ameralik (~30 g C m$^{-2}$ yr$^{-1}$ at AM10). Similarly, our results show that the OCBR at this very same station NK10 was about three times higher than at AM10. However, at basin scale, carbon burial remains similar in both fjords. These findings underscore the complexity of carbon burial dynamics in glacial fjords, highlighting that surface productivity and glacier type alone are not reliable predictors of OC burial.

**4.3 Pelagic and geomorphological influence on OC burial**

OC burial in fjord sediments is shaped not only by surface productivity but also by complex interactions between water column processes, fjord morphology, and bottom water conditions. There are several processes potentially at work leading to a decoupling of OC production in the water column and OC burial in the fjord sediments as discussed further and summarized in Fig 6.

**4.3.1 OC preservation conditions**

Since most of the OC deposited in both fjords is of marine origin, any differences in organic matter preservation between them are likely driven by environmental conditions rather than by differences in the nature of the organic material itself. The distinct geomorphology of Ameralik and Nuup Kangerlua, particularly their differing sill depths, likely shapes bottom water temperatures and may influence organic matter preservation within the fjords. Both

fjords have no anoxic deep water masses and bottom water renewal occurs every one to two years (Mortensen, 2011; Stuart-Lee et al., 2021), but bottom water temperature differs. Ameralik's shallower sill depth (~110 m) compared to Nuup Kangerlua (~200 m) restricts the inflow of warmer, saltier coastal waters (Stuart-Lee et al., 2021). Consequently, during field sampling, bottom water temperatures in Nuup Kangerlua were consistently warmer than in Ameralik, particularly in spring, with average values of 1.33 °C and 0.53 °C, respectively. The lower bottom water temperatures in Ameralik may explain the observed higher pigment and OC preservation in AM5 by reducing microbial degradation and slowing remineralization processes compared to sediments at the mouth of Nuup Kangerlua under influence of warmer waters. A comparative study of several Svalbard fjords suggested that relatively higher pigment content in sediments may be linked to lower bottom water temperatures (Krajewska et al., 2020). However, this hypothesis warrants further investigation, as Arctic microbial communities are well adapted to low temperatures, and mineralization rates below 10 °C appear to differ only minimally (Thamdrup et al., 2007; Scholze et al., 2020).

**4.3.2 Transport dynamics**

Besides potential differences in organic matter preservation, lateral transport may also influence the spatial distribution of OC across the seafloor. In Nuup Kangerlua, weak along-fjord gradients in sediment TOC, TN, and Chl-a content suggest dynamic currents that may redistribute OC. Estuarine and subglacial circulations, most active during melt season, can enhance OC export from inner to outer fjord (Mortensen et al., 2011; 2014; Juul-Pedersen et al., 2015).

At both fjord mouths, tidal mixing over sills drives baroclinic circulation, reintroducing nutrients into surface waters, promoting outer fjord surface productivity (Stuart-Lee et al., 2021, 2023). This aligns with higher TOC and pigment content as well as higher Chl-a:CPE ratios in Ameralik's outer fjord sediments. In contrast, Nuup Kangerlua sediments show no similar increase in TOC and pigment content in sediments of NK6 and NK7.

Sørensen et al. (2015) proposed that high POC export in Kobbefjord, a nearby glacier-free fjord, may result from OC input from Nuup Kangerlua. A similar OC transfer might explain a higher TOC and Chl-a content in the sediments toward Ameralik's mouth. While both fjords have estuarine and baroclinic circulation, stronger subglacial upwelling in Nuup Kangerlua likely enhances OC transport efficiency towards the fjord mouth. Ameralik may thus receive OC from outside, with deep basin retention supporting OC preservation (Fig. 6). The slightly coarser grain size at Ameralik's mid and outer stations, despite their distance from glacial input, may indeed reflect input from the entrance sill. Furthermore, the topography of Ameralik with the deep depression behind the sill can promote downslope transport and sediment accumulation, resulting in the relative higher TOC and pigment content at AM5 (Hargrave and Nielsen, 1976; Wassmann et al, 1984; Erlandsson, 2008). Therefore, hydrodynamics, downslope transport, or a combination of both can decouple surface productivity from local sediment deposition.

[Figure]

**Figure 6:** Schematic cross-sectional view of current regime and possible ways of phytoplankton or OC flow during summer in Nuup Kangerlua (**A**) and Ameralik (**B**) fjord systems. Tidal mixing above the sill area, estuarine circulation and intermediate baroclinic circulation occurs in both fjord systems, while the presence of MTGs in Nuup Kangerlua drives subglacial circulation through subglacial discharge. Nutrients are brought to the euphotic zone via tidal mixing and subglacial circulation. Turbid plumes, indicative of suspended sediment and organic matter input from glacier discharge and river runoff, are represented by the brown dotted pattern. Green arrows represent phytoplankton or OC transport and remineralization of organic carbon at the sediment-water

interface. A larger arrow points to higher expected flows. Station locations are marked along the fjords. The current dynamics illustrated for Nuup Kangerlua are based on Mortensen et al. (2018) and Stuart-Lee et al. (2023), while those for Ameralik are derived from Stuart-Lee et al. (2021; 2023).

**4.3.3 Food web OC uptake**

As both fjords exhibit a high contribution of marine-derived OC compared to other Arctic fjord systems (Fig. 5B), the unexpectedly higher sediment OC content in Ameralik's basin may reflect differences in carbon cycling pathways, both within sediments (stronger temperature-driven preservation, see 4.3.1) and in the overlying water column. In Nuup Kangerlua, greater phytoplankton biomass and a larger size class may support a more complex and efficient food web compared to Ameralik (Meire et al., 2023; Stuart-Lee et al., 2023), resulting in more OC being consumed or remineralized before it reaches the seafloor (Fig. 6). This is further supported by differences in zooplankton composition: Nuup Kangerlua hosts a higher proportion of large herbivorous copepods during the summer bloom, while smaller omnivorous taxa dominate in Ameralik (Stuart-Lee et al., 2024). However, despite these community differences, total zooplankton biomass did not differ significantly between fjords, possibly due to elevated predation pressure on larger zooplankton in Nuup Kangerlua (Stuart-Lee et al., 2024).

Elevated halibut landings in MTG-influenced fjords (Meire et al., 2017), combined with the known role of MTG fronts as productive foraging zones in Svalbard (Lydersen et al., 2014; Urbanski et al., 2017; Vacquié-Garcia et al., 2018; Hamilton et al., 2019), lend further support to the hypothesis that OC transfer through higher trophic levels is intensified in Nuup Kangerlua. This enhanced trophic transfer likely reduces vertical OC export, contributing to the lower sediment OC content observed despite higher pelagic productivity.

**4.4 Recommendations for future research**

The expected link between elevated surface primary production in MTG-influenced fjords and OCBR was not observed. Future studies should therefore examine the mechanisms controlling this mismatch between pelagic productivity and sediment burial. In addition, our results imply that glacial influence is not necessarily the most important factor steering OCBR, which means that more Greenland fjord systems should be studied to better understand the effect of retreating MTGs on OC burial. Based on our results we identified the following avenues for future research:

1. Mass accumulation rates and OCBRs need to be studied in Greenlandic and other Arctic fjords, ideally applying the CRS method, for standardized comparisons. As not all of our MARs could be determined by the CRS method, these estimates should be verified in the future.

2. Accurate carbon budget construction requires integrated knowledge of primary production, zooplankton grazing, pelagic and benthic biomass as well as pelagic and benthic mineralization rates (Spilling et al.,

2019), which are currently limited or lacking for these fjord systems. These parameters help quantify the mismatch between OC production and burial, which may arise from lateral transport processes or from OC incorporation into higher trophic levels. To address this, a more comprehensive understanding of food web dynamics and carbon flow in both fjords is essential.

3. An understanding of benthic OC cycling is important for quantifying carbon turnover at the sediment-water interface, potentially revealing processes that drive differences in OC burial efficiency in different fjord systems.

**5 Conclusion**

This study provides new insights into carbon burial processes in two southwest Greenland fjords with a different type of glacier influence. Our findings point to complex processes at work regarding carbon burial as our data revealed a different pattern than generally assumed in literature (Hopwood et al., 2020). Our data show that primary production generates most of the organic matter ending up at the seabed sediments in two sub-Arctic fjords with similar metamorphic and igneous catchment geology. Despite the upwelling mechanism in place sustaining more primary production, this process does not seem to induce a higher OC burial in the seabed sediments of a MTG-impacted fjord compared to a LTG-fed fjord. In contrast, this upwelling could be responsible for an export of carbon out the fjord or promoting the transfer of carbon through a more extensive food-web. In that case, MTGs could function as carbon pumps where an important part of the produced OC is stored beyond the fjord basin sediments. However, differences in geomorphology or bottom water characteristics between the two fjords can also override the importance of the subglacial nutrient supply and lead to a higher preservation of the OC in the fjord sediments.

Our findings highlight the importance of investigating both the pelagic as benthic compartment of Greenland fjord systems, which are understudied and underrepresented in global carbon budgets compared to other regions. Although this study advances our understanding of the carbon dynamics in Greenland fjords, several unresolved questions remain. For example, the role of physical circulation patterns in redistributing OC as well as differences in diagenetic processes between MTG- and LTG-influenced fjords, require further investigation. Additionally, the potential for complex food webs and more intense trophic interactions in MTG fjords to influence carbon sequestration deserves more attention.

Understanding the driving mechanisms of OCBR in fjord systems is essential to predict the impact of climate change on OC sequestration as MTGs evolve to LTGs. The similar OCBR observed between systems suggests that the retreat of MTGs from fjords may not necessarily reduce carbon burial, as new conditions influencing OCBR will emerge. Nevertheless, when assessing the impact of climate change on OC burial budgets, it is crucial to consider the fate of OC produced within the fjord.